# PathFinder: Graph-structured Reasoning for Medical Visual Question Answering

## Abstract

Medical Visual Question Answering (MVQA) aims not only to predict correct diagnoses but also to provide explicit, clinically-grounded reasoning to enhance interpretability, to foster clinician trust, and to support AI-assisted decision-making. Despite the recent advances, the explanations of existing MVQA are often incomplete and non-causal, neglecting key evidence, intermediate steps, and alternative hypotheses. In this paper, we present PathFinder, a graph-structured reasoning framework, in which medical entities are represented as nodes and causal/evidential relations as edges, enabling systematic traversal of diagnostic pathways. In PathFinder, we define two structural reasoning dimensions: step-wise exploration, which encourages PathFinder to traverse intermediate entities with causal links, and branch-wise exploration, which encourages exploring alternative diagnostic routes and ruling out unlikely options. Further, we introduce Graph-GRPO to integrate graph-structured supervision with two process-level rewards: a Step Reward for causally coherent reasoning, and a Branch Reward for systematic exploration of alternatives, complemented by outcome accuracy. Experiments on seven multimodal and seven text-only benchmarks consistently show that PathFinder outperforms state-of-the-art methods, while producing reasoning results that are causally coherent and structurally comprehensive. Codes will be released.

## 1 Introduction

*Reasoning-based* Vision-Language Models (VLMs) have recently emerged in Medical Visual Question Answering (MVQA) aiming not only to predict answers but also to generate explicit explanations that justify diagnostic decisions (Lai et al., 2025; Pan et al., 2025; Huang et al., 2025a; Sun et al., 2025a; Xu et al., 2025b; Chen et al., 2024b; Lin et al., 2025). Unlike methods that focus solely on producing correct answers, MVQA prioritize clinically-grounded reasoning, to improve interpretability, build clinician trust, and enable safer AI-assisted decision-making in clinical practice.

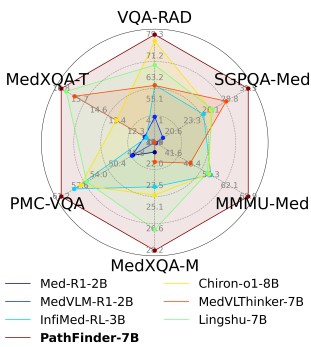

Figure 1: Comparing with six recent reasoning-based medical VLMs on six benchmarks. PathFinder-7B achieves the leading performance.

Despite these efforts, the quality of reasoning in practice often falls short. Even when the model outputs correct answers, their reasoning chains are frequently superficial or lack causal coherence. See two failure cases in Figure 2: MedVLM-R1 (Pan et al., 2025) and Med-R1 (Lai et al., 2025) select the correct answer but fail to specify any key imaging findings or supporting evidence. This may mislead surgeons, potentially causing them to overlook alternative procedures, thereby increasing the likelihood of intraoperative complications. Moreover, insufficient reasoning prevents clinicians from assessing answer reliability, ultimately jeopardizing patient safety (Cross et al., 2024; Xu et al., 2025a; Sokol et al., 2025; Challen et al., 2019). Therefore, answer correctness alone is insufficient in high-stakes medical settings: diagnostic reasoning should be *clinically-grounded*, integrating relevant evidence and systematically ruling out alternatives. In practice, reasoning rarely unfolds as a linear chain but instead forms a network of interdependent relations, where clinicians connect heterogeneous entities and weigh both supportive and contradictory evidence (Croskerry et al., 2023).

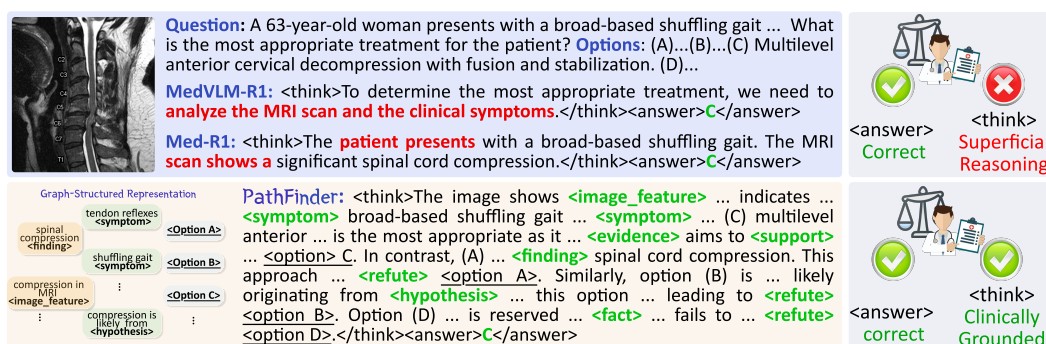

Figure 2: Examples of models choosing correct option but providing incomplete and non-causal reasoning without medical findings, imaging features or meaningful clues as intermediate steps. In contrast, **PathFinder-7B** arrives at the correct choice by traversing intermediate entities and integrating evidential and causal relations via its graph-structured representation.

Motivated by this observation, we introduce **PathFinder**, a graph-structured reasoning framework that links medical entities through evidential and causal relations to enable systematic exploration of diagnostic paths and construction of clinical reasoning chains. In PathFinder, reasoning is represented as a graph, with *nodes* corresponding to intermediate findings, symptoms, hypotheses, or conclusions, and *edges* representing causal or evidential links. This representation enables two complementary evaluation dimensions: (i) **Step-Wise Exploration**, whether reasoning progresses through intermediate entities and connects them causally; and (ii) **Branch-Wise Exploration**, whether alternative diagnostic routes are systematically considered and unlikely options eliminated. Evaluation of state-of-the-art general and medical VLMs reveals corresponding deficiencies: (i) *step deficiency*, reasoning chains are incomplete or non-causal, omitting key evidence; and (ii) *branch deficiency*, models produce narrow or overconfident predictions that fail to explore or eliminate alternative hypotheses. Figures 4 and 5 illustrate these issues, showing clinically shallow reasoning or premature commitment to a single option. More examples are in Appendix. A.2.

To address these deficiencies, we propose **Graph-GRPO**, to improve reasoning quality by integrating a graph-structured representation. Specifically, a **Step Reward** steers the model to generate complete and causally-coherent reasoning chains, while a **Branch Reward** promotes probing alternative hypotheses. These process-level rewards provide verifiable supervision for step-wise and branch-wise exploration. In addition, an outcome accuracy reward helps to ensure correct final predictions. As shown in Figure 1, our PathFinder achieves state-of-the-art accuracy comparing with recent reasoning-based medical VLM on both multi-modal benchmarks (VQA-RAD (Lau et al., 2018), PMC-VQA (Zhang et al., 2023), MMMU-Med (Yue et al., 2024), MedXQA-M (Zuo et al., 2025)) and tex-only benchmarks (SGPQA-Med (Team et al., 2025) and MedXQA-T (Zuo et al., 2025)), also generates richer, clinically-grounded reasoning, shown in Table 1.

Our contributions are threefold:

- We introduce PathFinder, a graph-structured reasoning framework that links medical entities via evidential and causal relations, enabling systematic construction of clinical reasoning chains and formalizes two complementary evaluation dimensions: step-wise and branch-wise exploration.

- We design Graph-GRPO, a reinforcement learning algorithm that leverages graph-structured rewards. Step Reward encourages causally coherent reasoning chains, while Branch Reward promotes exploration and elimination of alternatives.

- We develop PathFinder-7B and evaluate it on seven multimodal and seven text-only medical benchmarks. The model achieves state-of-the-art performance and significantly improves reasoning quality in both step- and branch-wise dimensions.

## 2 RELATED WORKS

**General Reasoning-based Vision Language Model.** Reasoning has become a central capability for large models, with significant progress made in both text-only and multimodal domains. Early

advances such as Chain-of-Thought (CoT) prompting (Wei et al., 2022) demonstrate that decomposing complex questions into intermediate steps improves logical consistency and problem-solving ability. Beyond supervised fine-tuning (SFT) on curated reasoning traces, reinforcement learning (RL) approaches have emerged as powerful alternatives, most notably Reinforcement Learning with Verifiable Rewards (RLVR), which directly optimizes answer correctness without requiring explicit reasoning supervision. Efficient RL algorithms such as Group Relative Policy Optimization (GRPO) (Shao et al., 2024) further enable scalable training without a critic network. Building on these ideas, models like DeepSeek-R1 (DeepSeek-AI et al., 2025) and Qwen-QwQ (Team, 2025) show that combining SFT with RL yields stronger reasoning skills, especially for long-context or multimodal tasks. Very recently, researchers (Wang et al., 2025; Yang et al., 2025; Zhang et al., 2025; Chen et al., 2025b) extend these pipelines to vision-language settings by first distilling reasoning traces from strong text-only LLMs then applying RL to improve visual reasoning.

**Chain-of-Thought Reasoning in Medical AI.** Chain-of-Thought (CoT) prompting has been widely used in general-domain LLMs to elicit interpretable intermediate reasoning. Its extension to the medical domain has recently drawn increasing attention, as clinical decision-making naturally follows multi-step diagnostic reasoning. Early works (Wu et al., 2024; Wei et al., 2024; Jiang et al., 2025; Liu et al., 2024; Gai et al., 2024) primarily explored CoT prompting for medical (V)QA, showing that exposing intermediate reasoning improves performance. These methods demonstrated that CoT can encourage models to articulate differential diagnoses or key clinical observations, but they often suffer from hallucinated steps, limited grounding, and a lack of evaluation on reasoning quality (Kim et al., 2025; Zuo & Jiang, 2024). More recently, reflective reasoning frameworks or multi-agent workflow, including MedPrompt (Chen et al., 2024c), MedReflect (Huang et al., 2025b) and MedReason (Sun et al., 2025b), enforce multi-step verification or reflection to partially mitigate overconfident clinical reasoning. However, these methods are still limited to linear reasoning chains and cannot capture the inherently branch and step-exploration nature of clinical workflows.

**Medical Reasoning-based Vision Language Model.** The development of medical vision-language models largely follows the trace of adapting general-purpose multimodal LLMs with domain-specific data. Early works such as Med-Flamingo (Moor et al., 2023), LLaVA-Med (Li et al., 2023), and RadFM (Wu et al., 2023) scaled training with large collections of medical image–text pairs, primarily focusing on improving representation alignment and clinical relevance. While these models achieve strong performance on image understanding and report generation, they are not designed for structured reasoning. Recently, there has been growing interest in equipping medical VLMs with reasoning capabilities. HuatuoGPT-o1 (Chen et al., 2024a) integrates reinforcement learning to elicit diagnostic reasoning. MedVLM-R1 (Pan et al., 2025) and Med-R1 (Lai et al., 2025) adopted reinforcement learning with verifiable rewards to improve reasoning quality across radiology and pathology tasks, albeit with limited data scales. More recent efforts such as GMAI-VL-R1 (Su et al., 2025), MedVLThinker (Huang et al., 2025a), Chiron-o1 (Sun et al., 2025a), InfiMed (Liu et al., 2025) and Lingshu (Xu et al., 2025b) extend this paradigm to broader multimodal medical reasoning. These studies highlight the emerging direction of integrating reasoning supervision and reinforcement learning into medical VLMs, marking an important step toward models capable of systematic diagnostic reasoning. However, the analysis in Table 1 reveals step and branch deficiencies in these prior medical VLMs, which skip critical intermediate reasoning steps or fail to explore alternative diagnostic routes. In comparison, our PathFinder framework explicitly addresses these deficiencies, yielding more comprehensive and clinically-grounded reasoning and setting a new state-of-the-art performance on multimodal and text-only benchmarks.

## 3 PATHFINDER: GRAPH-STRUCTURED REASONING FRAMEWORK

We introduce **PathFinder**, a graph-structured reasoning framework for MVQA. It consists of two core components: (a) **Graph-Structured Reasoning Representation** for CoT data construction, and (b) **Graph-GRPO** for reinforcement learning with two process-level rewards, Step Reward and Branch Reward, as well as outcome accuracy reward. This section is organized as follows: Section 3.1 defines two structural metrics for reasoning analysis; Section 3.2 details the graph-structured reasoning representation, and Section 3.3 introduces Graph-GRPO. Throughout this pipeline, we use OpenAI GPT-4o (Hurst et al., 2024) as an LLM-as-a-Judge for reasoning assessment and graph construction, and a licensed medical expert further manually verifies the generated outputs.

Table 1: Comparing with general- and medical-domain VLMs across four medical benchmarks. Each entry reports Accuracy (Acc., %), average Branch Exploration (BExp., %), and average Step Exploration (SExp.). Bold and underline values indicate the best and second-best accuracies among all compared models, while ***italic bold*** highlights our PathFinder-7B results. MedX-M and MedX-T denote MedXpertQA-MM and MedXpertQA-Text, respectively.

| Type | Method | MedX-M
Acc.↑/BExp.↑/SExp.↑ | MMMU-Med
Acc.↑/BExp.↑/SExp.↑ | MedX-T
Acc.↑/BExp.↑/SExp.↑ | MMLU-Pro
Acc.↑/BExp.↑/SExp.↑ |
|---|---|---|---|---|---|
| General | VL-Rethinker-7B | 22.6 / 81.7 / 8.43 | **62.8** / 70.1 / 6.71 | 13.4 / 78.6 / 12.53 | **56.1** / 85.9 / 10.95 |
| | R1-Onevision-7B | 21.1 / 82.9 / 10.69 | 55.2 / 71.1 / 8.57 | 10.7 / 78.8 / 16.11 | 42.1 / 83.4 / 13.70 |
| | R1-VL-7B | 20.9 / 62.4 / 6.21 | 57.2 / 48.0 / 4.99 | 11.1 / 47.8 / 7.57 | 44.3 / 45.6 / 5.87 |
| | VLAA-Thinker-7B | 23.0 / 59.0 / 7.05 | 56.6 / 54.6 / 5.08 | 12.5 / 37.3 / 7.05 | 50.0 / 56.4 / 5.08 |
| Medical | Med-R1-2B | 21.1 / 22.5 / 1.95 | 44.1 / 28.5 / 1.70 | 9.3 / 9.6 / 1.57 | 25.2 / 15.0 / 1.74 |
| | MedVLM-R1-2B | 19.1 / 22.7 / 2.09 | 42.1 / 28.1 / 2.39 | 9.0 / 9.4 / 1.50 | 24.8 / 16.5 / 1.92 |
| | Chiron-o1-8B | 24.6 / 69.3 / 8.32 | 48.3 / 57.8 / 5.92 | 13.5 / 58.8 / 10.85 | 47.2 / 63.9 / 9.68 |
| | Lingshu-7B | **25.1** / 48.1 / 5.77 | 54.0 / 57.7 / 5.46 | **15.3** / 27.1 / 5.66 | 50.2 / 54.1 / 6.74 |
| ***PathFinder-7B (Ours)*** | | ***28.2*** / 98.5 / 14.17 | ***68.9*** / 98.9 / 11.42 | ***16.8*** / 97.9 / 18.53 | ***55.4*** / 98.9 / 10.92 |

Table 2: Comparison of two reasoning structure metrics between accurate (Acc.) and failed cases for all 4 medical VLMs (Med-R1-2B, MedVLM-R1-2B, Chiron-o1-8B and Lingshu-7B) and 4 general VLMs (VL-Rethinker-7B, R1-Onevision-7B, R1-VL-7B and VLAA-Thinker-7B) across four datasets.

(a) Comparison of mean average BranchExploration.

| Type | | MedX-M | MMMU-Med | MedX-T | MMLU-Pro |
|---|---|---|---|---|---|
| General | ✓Acc. | 70.8 | 59.6 | 58.1 | 67.3 |
| | ✗ Failed | 71.7 | 62.8 | 61.0 | 68.3 |
| | ⇓ | *-0.9* | *-3.2* | *-2.9* | *-1.0* |
| Medical | ✓Acc. | 42.3 | 44.2 | 28.9 | 43.3 |
| | ✗ Failed | 40.2 | 41.8 | 25.8 | 33.9 |
| | ⇑ | *+2.1* | *+2.4* | *+3.1* | *+9.4* |

(b) Comparison of mean average StepExploration.

| Type | | MedX-M | MMMU-Med | MedX-T | MMLU-Pro |
|---|---|---|---|---|---|
| General | ✓Acc. | 7.90 | 6.28 | 10.72 | 9.10 |
| | ✗ Failed | 8.00 | 6.38 | 10.83 | 9.79 |
| | ⇓ | *-0.10* | *-0.10* | *-0.09* | *-0.69* |
| Medical | ✓Acc. | 4.82 | 4.01 | 5.69 | 5.92 |
| | ✗ Failed | 4.45 | 3.71 | 4.79 | 4.49 |
| | ⇑ | *+0.37* | *+0.30* | *+0.90* | *+1.43* |

## 3.1 STEP-WISE EXPLORATION & BRANCH-WISE EXPLORATION

To quantitatively evaluate the reasoning quality, we introduce two complementary structure metrics: (1) Step-Wise Exploration. We define StepExploration as the sum of number of reasoning steps per candidate option, reflecting whether the model provides layered and causally connected explanations rather than shallow justifications. (2) Branch-Wise Exploration. We denote BranchExploration as the fraction of candidate answer options for which the model produces a valid reasoning path. A valid path explicitly connects the question to the candidate option through a sequence of clinically meaningful intermediate steps.

$$\text{StepExploration} = \sum_{i=1}^{N_r} D_i, \ \text{BranchExploration} = \frac{\sum_{i=1}^{N_r} C_i}{N_r}, \ C_i = H(D_i) = \begin{cases} 1, & D_i > 0, \\ 0, & D_i = 0. \end{cases} \quad (1)$$

where $D_i$ denotes the number of reasoning step for each option, $N_r$ denotes the total number of candidate option, $C_i$ denotes if reasoning for option $i$ is valid with Heaviside function.

We evaluate four *medical VLMs* (Med-R1-2B (Lai et al., 2025), MedVLM-R1-2B (Pan et al., 2025), Chiron-o1-8B (Sun et al., 2025a) and Lingshu-7B (Xu et al., 2025b)) and four *general VLMs* (VL-Rethinker-7B (Wang et al., 2025), R1-Onevision-7B (Yang et al., 2025), R1-VL-7B (Zhang et al., 2025) and VLAA-Thinker-7B (Chen et al., 2025b)), all reasoning-oriented, across four benchmarks (two multimodal: MedXpertQA-MM (Zuo et al., 2025) and MMMU-Medical (Yue et al., 2024), and two text-only: MedXpertQA-Text (Zuo et al., 2025) and MMLU-Pro (Wang et al., 2024)). As shown in Table 1, medical VLMs often exhibit limited reasoning structure: their StepExploration and BranchExploration remain relatively low compared to general models, despite being specialized for the medical expertise. This indicates two prevalent issues: a *step-wise deficiency*, where models produce shallow or incomplete reasoning chains that may undermine the reliability of their diagnostic recommendations, and a *branch-wise deficiency*, where models fail to systematically explore all candidate options.

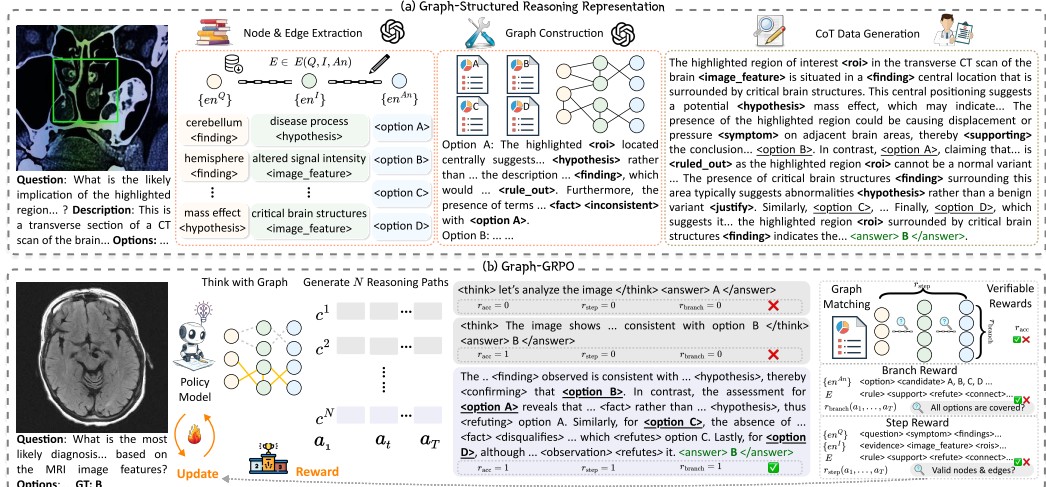

Figure 3: Overview of the proposed **PathFinder**. PathFinder consists of two key components: (a) **Graph-Structured Reasoning Representation**: given a medical VQA instance with image(s), question, and candidate options, we extract medical entities as nodes $\{en^Q, en^I, en^{An}\}$ and connect them via edges encoding causal or evidential relations $E \in E(Q, I, An)$. Multiple reasoning routes are constructed and pruned to form a coherent diagnostic reasoning graph, subsequently constructing graph-based CoT data for supervised cold-start. (b) **Graph-GRPO**: the constructed graph provides nodes and edges for reasoning path matching and generates structured process-level rewards ($r_{\text{step}}$ and $r_{\text{branch}}$) and outcome-level rewards ($r_{\text{acc}}$) for reinforcement learning.

To further investigate this effect, we compare these two exploration indicators between correct and incorrect cases in Table 2. The analysis reveals two important insights. (i) For medical VLMs, correct cases consistently exhibit higher step- and branch-wise exploration, suggesting that structured reasoning, when grounded in domain expertise, enhances performance. (ii) For general VLMs evaluated on medical datasets, the opposite trend emerges: incorrect cases tend to show greater reasoning complexity, possibly indicating that excessive or irrelevant reasoning may harm performance without sufficient medical knowledge. Overall, these findings highlight that *step- and branch-wise exploration within the reasoning structure serve as helpful indicators of improved performance for medical VLMs equipped with adequate domain knowledge.*

### 3.2 GRAPH-STRUCTURED REASONING REPRESENTATION

As shown in Figure 3(a), we construct a structured graph to represent diagnostic reasoning. This graph is built for a medical VQA pair consisting of medical image(s) with accompanying description $I$, a related question $Q$, a set of candidate options $An$, and the ground-truth answer $An_{gt} \in An$. First, we prompt the LLM to extract multiple sets of medical entities: $\{en^Q\}$ from the question, $\{en^I\}$ from the image, and $\{en^{An}\}$ from the candidate options. These entities are defined as *nodes*, which include *medical findings*, *clinical symptoms*, *hypotheses*, *rules*, *evidence*, and *imaging features* or *regions of interest*. Next, we establish *edges* to connect nodes. Each edge $E \in E(Q, I, An)$ encodes an inference operation such as *support*, *derive*, *refute*, or *rule out*, representing causal or evidential relationships among nodes. This design allows the reasoning graph to explicitly capture both confirmatory and eliminative diagnostic logic. For each candidate option $An_i \in An$, multiple reasoning paths may be generated that link $\{en^Q\}$ to $en^{An_i}$ through intermediate entities in $\{en^I\}$. To ensure conciseness and interpretability, we employ the LLM to prune irrelevant or redundant paths and retain a single coherent reasoning path for step-by-step generation. Finally, by repeating this process for all candidate options, we construct a complete graph-structured representation of the diagnostic reasoning process. This representation serves as the basis for subsequent reinforcement learning with Graph-GRPO. Detailed procedure for constructing the graph is provided in Appendix A.4.

## 3.3 GRAPH-GRPO

**Task Definition.** We formulate the reasoning process of a VLM as a sequential decision-making problem. Each medical VQA instance is denoted as $\{I, Q, An\}$, where $I$ represents the medical image(s), $Q$ is the clinical question and $An$ is the candidate options. The policy model $\pi_\theta$ generates a reasoning trajectory $c = (a_1, a_2, \ldots, a_T)$, where each action $a_t$ is sampled from $\pi_\theta$, corresponds to a reasoning unit, and $T$ is the maximum sequence length. The state $s_t$ denotes the partial reasoning chain generated up to step $t$, and updates as $s_{t+1} = (s_t, a_t)$. The overall objective is to optimize $\pi_\theta$ such that the generated trajectory is both clinically correct and structurally coherent. In reinforcement learning, this corresponds to maximizing the cumulative reward, where $r(s_t, a_t, s_{t+1})$, simplified as $r(a_t)$, denotes the reward for producing $a_t$ given $s_t$ and transitioning to $s_{t+1}$. Following prior works (Zhang et al., 2025; Yao et al., 2024), we define one action $a_t$ as generating a reasoning step, typically consisting of one or more sentences with multiple text tokens.

**Cold start via CoT Data SFT.** Before the reinforcement learning stage, we warm up the policy model $\pi_\theta$ through supervised fine-tuning with chain-of-thought (CoT) data. Importantly, the CoT data used here is not obtained by naive prompting, but is systematically derived from the graph-structured reasoning representation introduced in Section 3.2. Specifically, for each medical VQA pair, we leverage the constructed diagnostic graph to obtain reasoning trajectories: nodes and edges are unfolded into natural language explanations, forming coherent paths from question entities to candidate options. These structured routes provide high-quality CoT supervision that explicitly captures both the breadth of alternative branches and the depth of reasoning chains. The resulting trajectories are used as targets for maximum likelihood training, which encourages the policy to produce faithful, structured reasoning while also stabilizing subsequent reinforcement learning updates by initializing $\pi_\theta$ with clinically-grounded reasoning behaviors.

**Graph-based Group Relative Policy Optimization.** As illustrated in Figure 3(b), after cold start, we conduct reinforcement learning, which is built on the graph-structured reasoning representation. Each generated reasoning path of Graph-GRPO is mapped to the constructed diagnostic graph, enabling the computation of two complementary process-level rewards $r_{\text{step}}$ and $r_{\text{branch}}$. Unlike sparse accuracy rewards, these process-level rewards offer dense supervision at the level of individual reasoning steps, encouraging both step-wise logical coherence and route-wise comprehensiveness.

(1) **Step Reward** ($r_{\text{step}}$) enforces a *step-wise, clinically coherent reasoning process*. At each reasoning step, the trajectory is rewarded only if the action $a_t$ correctly incorporates relevant intermediate entities from the graph–$\{en^Q\}$ from the question and $\{en^I\}$ from the image–and links them through the appropriate causal or evidential edges $E$. We formalize this as: at each step, a reward is granted only when the action $a_t$ includes entities nodes $\{en^Q\} \cup \{en^I\}$ and links them correctly according to the graph edges $E(Q, I, An)$. $r_{\text{step}}$ is defined as $r_{\text{step}}(a_1, \ldots, a_T) = \frac{1}{T} \sum_{t=1}^{T} r_{\text{step}}(a_t)$, where

$$r_{\text{step}}(a_t) = \begin{cases} 1, & \text{if } N(a_t) \subset \{en^Q\} \cup \{en^I\} \text{ and } E(a_t) \subset E(Q, I, An), \\ 0, & \text{otherwise.} \end{cases} \quad (2)$$

here $N(a_t)$ and $E(a_t)$ denote the set of entities and edges included in action $a_t$.

(2) **Branch Reward** ($r_{\text{branch}}$) enforces *branch-wise exploration*, ensuring that the reasoning trajectory does not tunnel toward a single candidate option but systematically considers all candidate options $An$. Formally, $r_{\text{branch}}$ is defined as the fraction of candidate options that are explicitly examined during reasoning. A candidate $An_i$ is counted as covered if it appears in at least one reasoning step $a_\tau$. $r_{\text{branch}}$ is formularized as:

$$r_{\text{branch}}(a_1, \ldots, a_T) = \frac{1}{|An|} \sum_i \mathbf{1}\left[\exists a_\tau \text{ s.t. } en^{An_i} \in a_\tau\right] \quad (3)$$

The overall reward integrates both process-level and outcome-level signals. At the process level, we combine the above two dimensions, ensuring that the learned policy produces reasoning chains that are simultaneously step-wise coherent and branch-wise comprehensive. At the outcome level, an accuracy reward enforces correctness of the final prediction. Formally, the process-level reward for one generated reasoning path is defined as:

$$r_{\text{proc}} = \lambda_{\text{step}} \, r_{\text{step}}(a_1, \ldots, a_T) + \lambda_{\text{branch}} \, r_{\text{branch}}(a_1, a_2, \ldots, a_T), \quad (4)$$

Table 3: Comprehensive evaluation on medical multimodal benchmarks. **Bold** and underline scores indicate the best and the second-best performance excluding close-source models. OMVQA and MedXQA represent OmniMedVQA and MedXpertQA-MM benchmarks.

| Models | Out-of-Domain | | In-Domain | | | | |
|---|---|---|---|---|---|---|---|
| | MedXQA↑ | MMMU-Med↑ | VQA-RAD↑ | PMC-VQA↑ | PathVQA↑ | SLAKE↑ | OMVQA↑ |
| *Close-source proprietary models* | | | | | | | |
| GPT-4.1 | 45.2 | 75.2 | 65.0 | 55.2 | 55.5 | 72.2 | 75.5 |
| Claude Sonnet | 43.3 | 74.6 | 67.6 | 54.4 | 54.2 | 70.6 | 65.5 |
| Gemini-2.5-Flash | 52.8 | 76.9 | 68.5 | 55.4 | 55.4 | 75.8 | 71.0 |
| *Open-source non-reasoning models* | | | | | | | |
| BiomedGPT | - | 24.9 | 16.6 | 27.6 | 11.3 | 13.6 | 27.9 |
| MedGemma-4B | 22.3 | 43.7 | 72.5 | 49.9 | 48.8 | 76.4 | 69.8 |
| LLaVA-Med-7B | 20.3 | 29.3 | 53.7 | 30.5 | 38.8 | 48.0 | 44.3 |
| HuatuoGPT-V-7B | 21.6 | 47.3 | 67.0 | 53.3 | 48.0 | 67.8 | 74.2 |
| BioMediX2-8B | 21.8 | 39.8 | 49.2 | 43.5 | 37.0 | 57.7 | 63.3 |
| *Open-source reasoning-based models* | | | | | | | |
| Med-R1-2B | 21.1 | 34.8 | 39.0 | 47.4 | 15.3 | 54.5 | 69.9 |
| MedVLM-R1-2B | 20.4 | 35.2 | 48.6 | 47.6 | 32.5 | 56.0 | 77.7 |
| InfiMed-RL-3B | 23.6 | 55.3 | 60.5 | 58.7 | 62.0 | 82.4 | 71.7 |
| GMAI-VL-R1-7B | 23.8 | 57.3 | - | - | - | - | 61.0 |
| Lingshu-7B | 26.7 | 54.0 | 67.9 | 56.3 | 61.9 | 83.1 | **82.9** |
| Chiron-o1-8B | 24.2 | 54.6 | 76.8 | 57.5 | 74.0 | **83.2** | 65.7 |
| MedVLThinker-7B | 21.8 | 47.8 | 60.3 | 43.2 | 51.4 | 55.2 | 62.1 |
| *PathFinder-7B (Ours)* | **28.2** | **68.9** | **79.3** | **61.2** | **87.0** | 80.3 | 63.5 |

where $\lambda_{\text{step}}, \lambda_{\text{branch}} \geq 0$ are tunable coefficients that balance the two objectives. The final outcome reward is given by: $r_{\text{acc}}^i(y, \hat{y}) = \mathbb{1}[\hat{y} = y]$, where $y$ is the ground-truth answer, $\hat{y}$ is the model's final prediction, and $\mathbb{1}[\cdot]$ is the indicator function. Accordingly, the overall reward for a trajectory is:

$$r = r_{\text{proc}} + \lambda_{\text{acc}} r_{\text{acc}}(y, \hat{y}) = \lambda_{\text{step}} r_{\text{step}}(a_1, \ldots, a_T) + \lambda_{\text{branch}} r_{\text{branch}}(a_1, a_2, ..., a_T), + \lambda_{\text{acc}} r_{\text{acc}}(y, \hat{y}) \tag{5}$$

where $\lambda_{\text{acc}}$ controls the relative weight of the final accuracy reward. Finally, GRPO normalizes the rewards by using their mean and standard deviation, and computes the advantage as:

$$A^i = \frac{r^i - \text{mean}\{r^1, r^2, ..., r^N\}}{\text{std}\{r^1, r^2, ..., r^N\}} \tag{6}$$

where $A^i$ with $i \subseteq [1, N]$ represents the advantage of the $i$-th candidate reasoning trajectory $c^i = (a_1^i, a_2^i, ..., a_T^i)$. Here, $N$ represents total number candidate reasoning trajectory.

# 4 EXPERIMENT

## 4.1 EXPERIMENTAL SETUP

**Data Source.** To support both cold start initialization and Graph-GRPO, we leverage a diverse collection of multimodal and text-only medical datasets. For the cold start stage, we construct around 100k chain-of-thought data via graph-structured reasoning representation strategy from both multimodal and text-only data sources. For multimodal data, we sample from PubMedVision (Chen et al., 2024b), MedTrinity-25M (Xie et al., 2025), and GMAI-Reasoning10K-SFT (Su et al., 2025). For the text-only data, we sample from Medical23k (Huang et al., 2025a), medical-o1-reasoning-SFT (Chen et al., 2024a), Medical-R1-Distill-Data (Chen et al., 2024a), and MedReason (Wu et al., 2025). For the Graph-GRPO stage, we sample training data from VQA-RAD (Lau et al., 2018), PathVQA (He et al., 2020), PMC-VQA (Zhang et al., 2023), SLAKE (Liu et al., 2021) and GMAI-Reasoning10K-RL (Su et al., 2025) as multimodal data, and from MedAgentsBench (Tang et al., 2025) as text-only data, yielding around 14k samples in total. Detailed data distribution is described in Appendix. A.4.

**Implementation Details.** We use the state-of-the-art open-source VLM, Qwen2.5-VL-7B-Instruct (Bai et al., 2025) as baseline. For cold start, we set the batch size of 4 per device and learning rate of $2e^{-5}$ with cosine decay scheduler to train 2 epochs. For Graph-GRPO, we set the number of generation per sample $N = 4$ with batch size of 4 per device, $\lambda_{\text{depth}} = \lambda_{\text{breadth}} = 0.5$ and $\lambda_{\text{final}} = 1$

with learning rate of $1e^{-5}$ to train 1 epoch. For both stages, we freeze the vision encoder and vision-language merger for finetuning. All experiments are conducted on 4×A6000-48GB GPUs.

Table 4: Comprehensive evaluation on medical text-only benchmarks. **Bold** and underline scores indicate the best and the second-best performance excluding close-source models. MedXQA and SGPQA-Med represents MedXpertQA-Text and SuperGPQA medicine discipline benchmark.

| Models | Out-of-Domain | | In-Domain | | | | |
|---|---|---|---|---|---|---|---|
| | MedXQA↑ | SGPQA-Med↑ | MMLU↑ | PubMedQA↑ | MedMCQA↑ | MedQA↑ | Medbullets↑ |
| *Close-source proprietary models* | | | | | | | |
| GPT-4.1 | 30.9 | 49.9 | 89.6 | 75.6 | 77.7 | 89.1 | 77.0 |
| Claude Sonnet | 33.6 | 56.3 | 91.3 | 78.6 | 79.3 | 92.1 | 80.2 |
| Gemini-2.5-Flash | 35.6 | 53.3 | 84.2 | 73.8 | 73.6 | 91.2 | 77.6 |
| *Open-source non-reasoning models* | | | | | | | |
| MedGemma-4B | 12.8 | 21.6 | 66.7 | 72.2 | 52.2 | 56.2 | 45.6 |
| LLaVA-Med-7B | 9.9 | 16.1 | 50.6 | 26.4 | 39.4 | 42.0 | 34.4 |
| HuatuoGPT-V-7B | 10.1 | 21.9 | 69.3 | 72.8 | 51.2 | 52.9 | 40.9 |
| BioMediX2-8B | 13.4 | 25.2 | 68.6 | 75.2 | 52.9 | 58.9 | 45.9 |
| *Open-source reasoning-based models* | | | | | | | |
| Med-R1-2B | 11.2 | 17.9 | 51.5 | 66.2 | 39.1 | 39.9 | 33.6 |
| MedVLM-R1-2B | 11.8 | 19.1 | 51.8 | 66.4 | 39.7 | 42.3 | 33.8 |
| InfiMed-RL-3B | 11.7 | 25.0 | 68.8 | 75.0 | 50.5 | 53.5 | 40.3 |
| Lingshu-7B | 16.5 | 26.3 | 74.5 | 76.6 | 55.9 | 63.3 | **56.2** |
| Chiron-o1-8B | 13.5 | 26.1 | 68.5 | 70.0 | 51.5 | 49.7 | 38.6 |
| MedVLThinker-7B | 16.0 | 28.3 | 76.4 | 65.0 | 52.1 | 60.3 | 44.8 |
| *PathFinder-7B (Ours)* | **16.8** | **31.5** | **80.2** | **77.2** | **56.8** | **66.3** | 46.4 |

## 4.2 EXPERIMENTAL RESULTS

To comprehensively evaluate our proposed **PathFinder**, we compare it with state-of-the-art VLMs, including close-source proprietary models, open-source non-reasoning and reasoning-based models, across 7 multimodal and 7 text-only benchmarks on leaderboards (Xu et al., 2025b). For multimodal evaluation, MedXpertQA-MM (Zuo et al., 2025) and MMMU Medical validation (Yue et al., 2024) represents out-of-domain benchmarks, while VQA-RAD (Lau et al., 2018), PathVQA (He et al., 2020), PMC-VQA (Zhang et al., 2023), SLAKE (Liu et al., 2021) and OmniMedVQA (Hu et al., 2024) are in-domain benchmarks since in Graph-GRPO, we include training samples from VQA-RAD, PathVQA and PMC-VQA, with GMAI-Reasoning10K sharing the same curated dataset distribution with OmniMedVQA. For text-only evaluation, MedXpertQA-Text (Zuo et al., 2025) and SuperGPQA medicine (Team et al., 2025) represent out-of-domain benchmarks, whereas MMLU (Hendrycks et al., 2020), PubMedQA (Jin et al., 2019), MedMCQA (Pal et al., 2022), MedQA (Jin et al., 2021) and MedBullets (Chen et al., 2025a) are in-domain benchmarks since MedAgentsBench and MedReason are sampled from these datasets. Open-source models are smaller than 10B.

**Multimodal Results** in Table 3. For BiomedGPT (Zhang et al., 2024), we note that it does not support multi-image input of MedXpertQA-MM. For MedVLThinker-7B (Huang et al., 2025a), we select its model which follows the SFT+RL training paradigm. We include the reported results from Chiron-o1-8B (Sun et al., 2025a) and InfiMed-RL-3B (Liu et al., 2025). As GMAI-VL-R1-7B (Su et al., 2025) has not yet released its model, we only compare its reported numbers. Overall, our proposed PathFinder-7B demonstrates **state-of-the-art** performance across both out-of-domain and in-domain benchmarks. In particular, it achieves the *best results* among open-source models on the out-of-domain expert-level understanding and reasoning datasets, MedXpertQA-MM and MMMU-Medical, surpassing the previous model, Lingshu-7B and Chiron-o1-8B, by a substantial margin (28.2 vs. 26.7 on MedXQA and 68.9 vs. 54.6 on MMMU-Med). On in-domain datasets, PathFinder-7B also attains top performance on several benchmarks, including VQA-RAD, PMC-VQA, and PathVQA, demonstrating its ability to integrate multimodal medical evidence for accurate answer and clinically-grounded reasoning. While Chiron-o1-8B achieves the highest score on SLAKE, PathFinder maintains overall superior reasoning performance across datasets. On OMVQA, PathFinder-7B achieves 63.5, which is lower than the best model, Lingshu-7B (82.9). This is primarily because PathFinder-7B's in-domain training data is derived from GMAI-Reasoning10K, which shares the same curated dataset distribution with OMVQA. Notably, PathFinder-7B still surpasses its source model GMAI-VL-R1 (from 61.0 to 63.5), indicating that

our graph-structured reasoning and Graph-GRPO meaningfully improves the performance beyond the original training data.

**Text-only Results** in Table 4. As Chiron-o1-8B (Sun et al., 2025a) only report the results of MedXpertQA-Text, we inference the model to obtain the rest of results following its official settings. Similarly, we infer the InfiMed-RL-3B and MedVLThinker-7B to produce the results. Our proposed PathFinder-7B achieves **state-of-the-art** performance among open-source models across both out-of-domain and in-domain text-only benchmarks. On out-of-domain datasets, it surpasses previous best models on MedXQA (16.8 vs. 16.5 for Lingshu-7B) and SGPQA-Med (31.5 vs. 28.3 for MedVLThinker-7B). For in-domain benchmarks, PathFinder-7B also attains top performance on MMLU, PubMedQA, MedMCQA, and MedQA, demonstrating its ability to leverage structured reasoning over textual medical evidence. While Lingshu-7B achieves the highest score on Medbullets (56.2), PathFinder-7B maintains the second-best performance (46.4).

**More than Accurate.** Prior medical VLMs primarily optimize on accuracy, they often exhibit *step-deficiency* and *branch-deficiency*, failing to fully explore intermediate reasoning steps or alternative diagnostic routes. As shown in Table 1, although Chiron-o1-8B and Lingshu-7B achieve competitive accuracies (24.6 and 25.1 on MedX-M; 13.5 and 15.3 on MedX-T), their step-wise exploration (DExp.) and branch-wise exploration (BExp.) remain limited (e.g., $\leq 69.3\%$ BExp. and $\leq 10.85$ DExp.). In contrast, PathFinder-7B, optimized via Graph-GRPO, not only attains the highest accuracy across benchmarks (e.g., 28.2 on MedX-M and 16.8 on MedX-T) but also demonstrates substantially improved reasoning quality, achieving comprehensive branch-wise exploration ($\sim$98–99%) and significantly deeper step-wise reasoning chains (14.17 and 18.53). This indicates that Graph-GRPO effectively mitigates step and branch deficiency, guiding the model toward more comprehensive, and grounded reasoning rather than relying on shallow or narrow solution paths. Examples are illustrated in Appendix A.9.

### 4.3 ABLATION STUDY

We perform ablation studies on Qwen2.5-VL-7B to evaluate the contribution of each component in our PathFinder framework on both multimodal and text-only benchmarks. Table 5 summarizes the results. (1) **Effectiveness of $r_{acc}$ without cold-start:** comparing the first and third rows, we observe that applying accuracy-based reinforcement learning alone, without cold-start initialization of explicit medical knowledge, yields minimal improvements. (2) **Impact of cold-start initialization via graph-structured representation CoT data:** adding cold-start initialization substantially boosts performance (from the first to the second row), confirming that explicit domain knowledge is crucial to guide the model toward medically relevant reasoning trajectories. (3) **Effectiveness of $r_{acc}$ with cold start:** introducing accuracy reward on top of cold start initialization further improves performance (from the second to the fourth row), showing that reinforcement learning becomes more effective once the model is properly initialized. (4) **Effectiveness of structural rewards:** incorporating step-wise ($r_{st.}$) or branch-wise ($r_{br.}$) rewards leads to additional improvements, with each dimension contributing progressively. (5) **PathFinder:** combining cold start initialization, accuracy reward, and two structural rewards achieves the best overall performance across all benchmarks, validating the synergistic effect of accuracy-driven optimization and structured reasoning guidance. More ablation results are provided in Appendix. A.7.

Table 5: Ablation study of different components in PathFinder.

| Training Strategy | | | | Multi-Modal | | Text-Only | |
|---|---|---|---|---|---|---|---|
| init. | $r_{acc}$ | $r_{st.}$ | $r_{br.}$ | MedX-M↑ | VQA-RAD↑ | MedX-T↑ | MMLU↑ |
| | | | | 22.2 | 64.5 | 12.9 | 73.4 |
| ✓ | | | | 26.1 | 77.1 | 15.0 | 78.1 |
| | ✓ | | | 22.7 | 65.2 | 13.0 | 73.8 |
| ✓ | ✓ | | | 27.4 | 78.5 | 16.1 | 79.1 |
| ✓ | ✓ | ✓ | | 28.0 | 78.8 | 16.3 | 79.7 |
| ✓ | ✓ | | ✓ | 27.9 | 79.0 | 16.5 | 79.4 |
| ✓ | ✓ | ✓ | ✓ | 28.2 | 79.3 | 16.8 | 80.2 |

### 5 CONCLUSION

In this work, we tackle two persistent reasoning deficiencies in medical vision–language models (VLMs): step-wise deficiency, where reasoning chains are incomplete or non-causal, and branch-wise deficiency, where models fail to systematically explore alternative diagnostic routes. To address these limitations, we propose PathFinder, a graph-structured reasoning framework that represents

diagnostic processes as interconnected medical entities with causal and evidential links. To encourage both step- and branch-wise exploration, we introduce Graph-GRPO, which leverages graph-structured supervision and incorporates Step Reward, Branch Reward, and outcome accuracy. Step Reward encourages causally coherent, layered reasoning, while Branch Reward ensures thorough consideration of alternative diagnostic paths. Extensive experiments on seven multimodal and seven text-only medical benchmarks demonstrate that our PathFinder not only achieves state-of-the-art accuracy but also produces reasoning that is deeper, more comprehensive, and clinically-grounded. Overall, PathFinder with Graph-GRPO provides a principled framework for verifiable and trustworthy diagnostic reasoning in medical VLMs.

**Reproducibility Statement.** Codes in this work will be publicly released to facilitate reproducibility. The graph-structured reasoning framework, PathFinder, and Graph-GRPO, are described in Section 3 and 3.3, including architecture details, reward definitions, and training procedures. Appendix A.4 provides a detailed description of dataset construction and preprocessing for both multimodal and text-only medical data source. Prompts for constructing the graph-structured chain-of-thought (CoT) data are also provided in Appendix A.4. All experimental configurations, hyper-parameters are reported in the main text and Appendix. Together, these resources ensure that the results reported in this paper can be reproduced.

**Ethics Statement.** This work adheres to the ICLR Code of Ethics. Our study does not involve direct interaction with patients or the collection of personally identifiable medical data. All datasets used are either publicly available under appropriate research licenses or constructed from de-identified resources (e.g., PubMedVision (Chen et al., 2024b), MedTrinity-25M (Xie et al., 2025), GMAI-Reasoning10k Su et al. (2025), Medical23k Huang et al. (2025a), medical-o1-reasoning-SFT Chen et al. (2024a), Medical-R1-Distill-Data (Chen et al., 2024a), MedReason (Wu et al., 2025), VQA-RAD (Lau et al., 2018), PathVQA (He et al., 2020), PMC-VQA (Zhang et al., 2023), SLAKE (Liu et al., 2021), MedAgentsBench (Tang et al., 2025), MedXperQA (Zuo et al., 2025), MMMU Medical Validation (Yue et al., 2024), OmniMedVQA (Hu et al., 2024), SuperGPQA (Team et al., 2025), MMLU (Hendrycks et al., 2020), MMLU-Pro (Wang et al., 2024), PubMedQA (Jin et al., 2019), MedMCQA (Pal et al., 2022), MedQA (Jin et al., 2021), MedBullets Chen et al. (2025a)). We ensure that no private, sensitive or confidential health information is included. The reasoning data constructed with GPT-4o is further verified by licensed medical experts to mitigate potential errors or misleading medical content. We acknowledge that medical AI research carries potential risks if misapplied. To reduce such risks, our work is intended solely for research purposes and should not be used for clinical decision-making without human oversight. We highlight the limitations of our approach in requiring extensive domain knowledge and stress the need for careful expert validation before any downstream deployment. Code, data construction procedures, and prompts are transparently documented (Appendix A.4 and A.4) to promote reproducibility and responsible use.

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

# A APPENDIX

## A.1 LLM USAGE DISCLOSURE

Large Language Models (LLMs), specifically OpenAI ChatGPT [1], were employed in several auxiliary roles during the preparation of this manuscript. First, LLMs were used for writing assistance, including language polishing, grammar correction, and improving the overall readability of the manuscript. Second, LLMs were utilized as *LLM-as-a-Judge* to assist in the evaluation process, particularly in accessing model outputs for evaluation. Third, LLMs were incorporated into parts of the data construction pipeline, where they are used for our Graph-based reasoning representation and CoT data construction. Finally, LLMs were leveraged to assist in case study analysis, helping to structure observations and highlight key reasoning patterns. LLMs were not involved in the conception of research ideas or in the design of the proposed methodology implementation. The authors take full responsibility for all research findings and the content of this paper.

## A.2 FAILURE CASE STUDY

In this paper, all the example cases are inferred following the official settings of individual repo. We provide failed case of Med-R1 (Lai et al., 2025), MedVLM-R1 (Pan et al., 2025), Chiron-o1 (Sun et al., 2025a) and Lingshu (Xu et al., 2025b) on MedXpertQA-MM (Zuo et al., 2025) and MMMU Medical Validation in (Chen et al., 2024b) in Figure 4, 5, 6 and 7.

In Figure 4, cases (a–f) illustrate the problem of *step-wise deficiency*, where models produce reasoning chains that appear superficially coherent but fail to capture the full causal or evidential progression. For instance, in (a) the patient with ischemic toe discoloration and an abdominal aortic aneurysm, the model chose "calcific sclerosis," overlooking the embolic mechanism of mural thrombus generating fibrinous fragments that lodge in distal arteries. The absence of a pathophysiologic bridge from aneurysm to digital ischemia exemplifies shallow reasoning. In (b), the chronic finger injury with failed splinting was attributed to postoperative stiffness, while the correct answer required recognizing hardware prominence as the clinically significant complication; here, the model truncated its analysis to joint mobility without incorporating radiographic evidence of fixation hardware. Similarly, (c) involved integrating T1-weighted MRI with FDG PET; the model incorrectly chose metastatic thymoma, reflecting failure to link metabolic homogeneity and anatomic mediastinal involvement toward lymphoma. In (d), acute intoxication with abdominal tenderness was misclassified as septic shock, revealing the model's tendency to stop at surface-level vital instability rather than connecting alcohol use, abdominal pain, and thrombocytopenia to pancreatitis. In (e), the musculoskeletal lesion was labeled as PVNS, but the correct diagnosis of rheumatoid arthritis required reasoning over chronic inflammatory features that were ignored. Finally, (f) demonstrates symbolic grounding gaps: the model recognized circular DNA as mitochondrial but mismapped the labeled diagram, exposing an incomplete linkage between genetic reasoning and visual grounding. Collectively, these failures reveal step deficiency as an inability to sustain multi-step, evidence-linked causal reasoning.

Additionally, cases (g–l) in Figure 5 highlight *branch deficiency*, where models prematurely narrow diagnostic space and neglect to compare against more plausible alternatives. In (g), the patient with necrotizing soft tissue infection was attributed to Vibrio vulnificus, an exotic pathogen, while ignoring the overwhelmingly likely cause of group A streptococcus; the model failed to weigh prevalence and exposure history in its reasoning. In (h), fibular plating was described as reducing hardware prominence, but the correct consideration was increased intra-articular screw penetration risk; the model privileged a single surface-level benefit without systematically evaluating trade-offs. In (i), a case of varicocele was incorrectly tied to infection (positive chlamydia NAAT) rather than its established association with impaired semen quality, underscoring insufficient exploration of canonical sequelae. Similarly, in (j), a testicular mass complication was predicted as abscess rather than atrophy, missing the broader range of longitudinal consequences. In (k), when selecting a biomarker for endocrine tumor, the model chose $\beta$2-microglobulin, a nonspecific marker, while neglecting the disease-defining calcitonin—again narrowing prematurely instead of comparing discriminative markers. Finally, in (l), orbital swelling was judged least likely due to oral contraceptives, yet the more appropriate elimination was dermatomyositis; here, the model overemphasized trivial exclu-

---

[1] https://chat.openai.com/chat

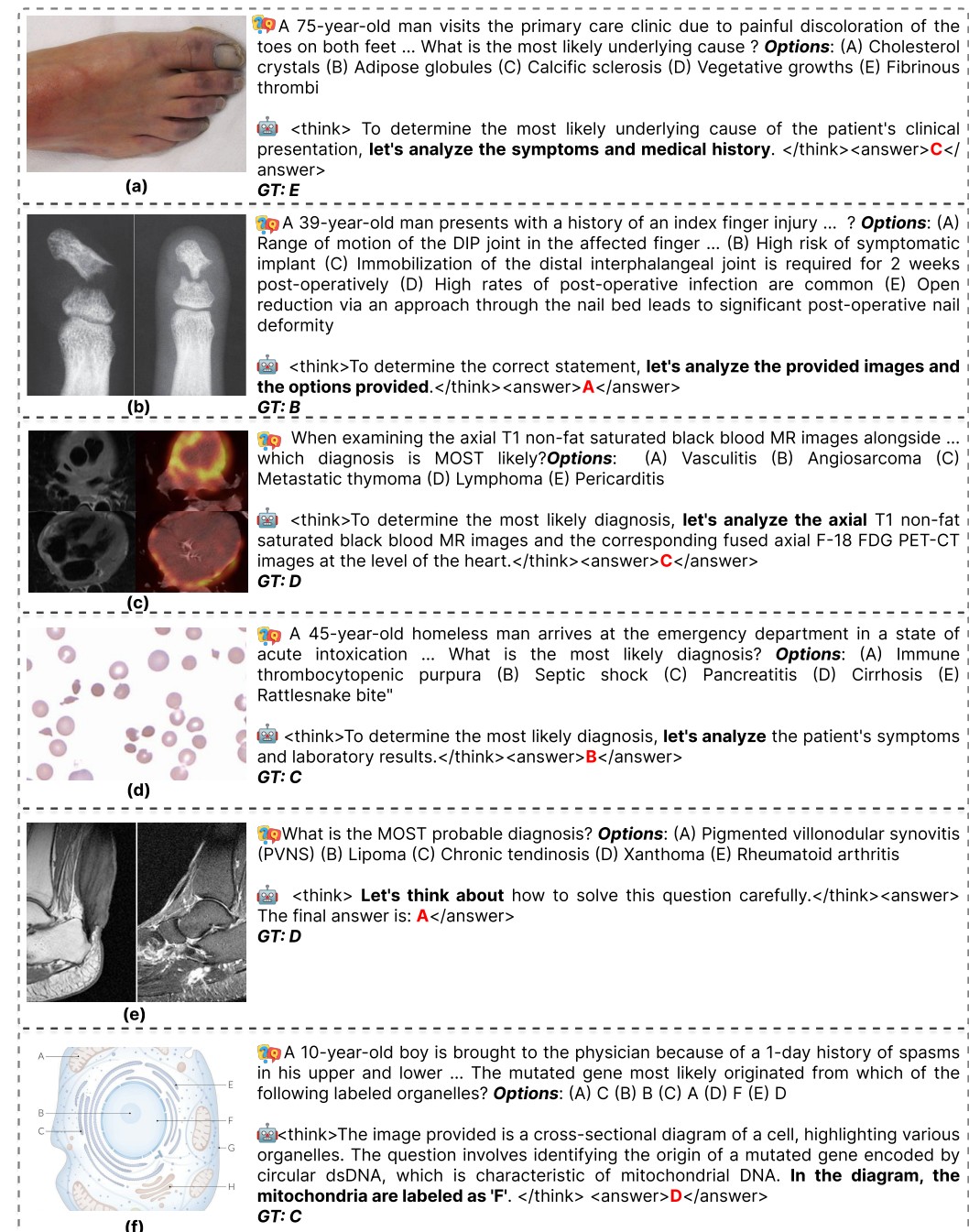

Figure 4: Failure case of *step-wise deficiency*(a)(b)(c)(d)(e)(f) of Med-R1 (Lai et al., 2025), MedVLM-R1 (Pan et al., 2025), Chiron-o1 (Sun et al., 2025a) and Lingshu (Xu et al., 2025b) on MedXpertQA-MM.

sion while failing to reject a clinically inconsistent autoimmune cause. These branch errors illustrate a tendency to commit early to plausible but incomplete explanations, without maintaining wide-enough diagnostic coverage. Together, step and branch deficiencies show complementary failure modes: the former yields shallow causal chains, while the latter collapses the hypothesis space too early both undermining safe and reliable clinical reasoning.

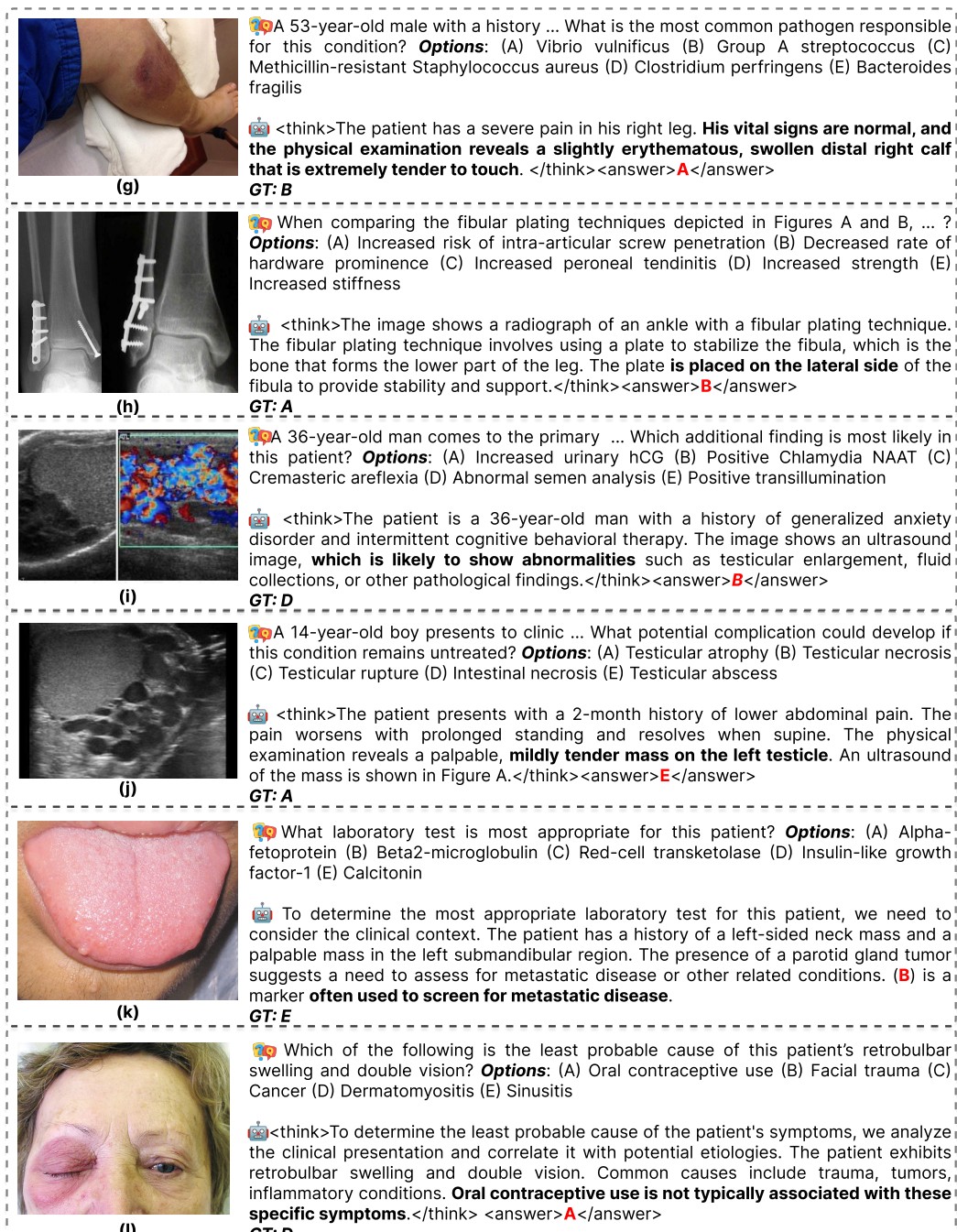

Figure 5: Failure case of *branch deficiency*(g)(h)(i)(j)(k)(l) of Med-R1 (Lai et al., 2025), MedVLM-R1 (Pan et al., 2025), Chiron-o1 (Sun et al., 2025a) and Lingshu (Xu et al., 2025b) on MedXpertQA-MM.

In Figure 6, (a) in abdominal radiography, the model incorrectly classifies a lateral decubitus x-ray as an upright abdominal film, overlooking positional cues in the imaging context. (b) For cerebrovascular supply, the model wrongly selects bilateral MCA rather than the left MCA, reflecting a failure to integrate the language-dominant hemisphere in speech. (c) In neuroanatomy, although the model answers correctly about the tuberoinfundibular tract, the reasoning provided is generic and does not specify its neuroendocrine role, illustrating shallow explanation. (d) For cerebellar lesions, the model correctly chooses "all are correct" but provides no justification, skipping over

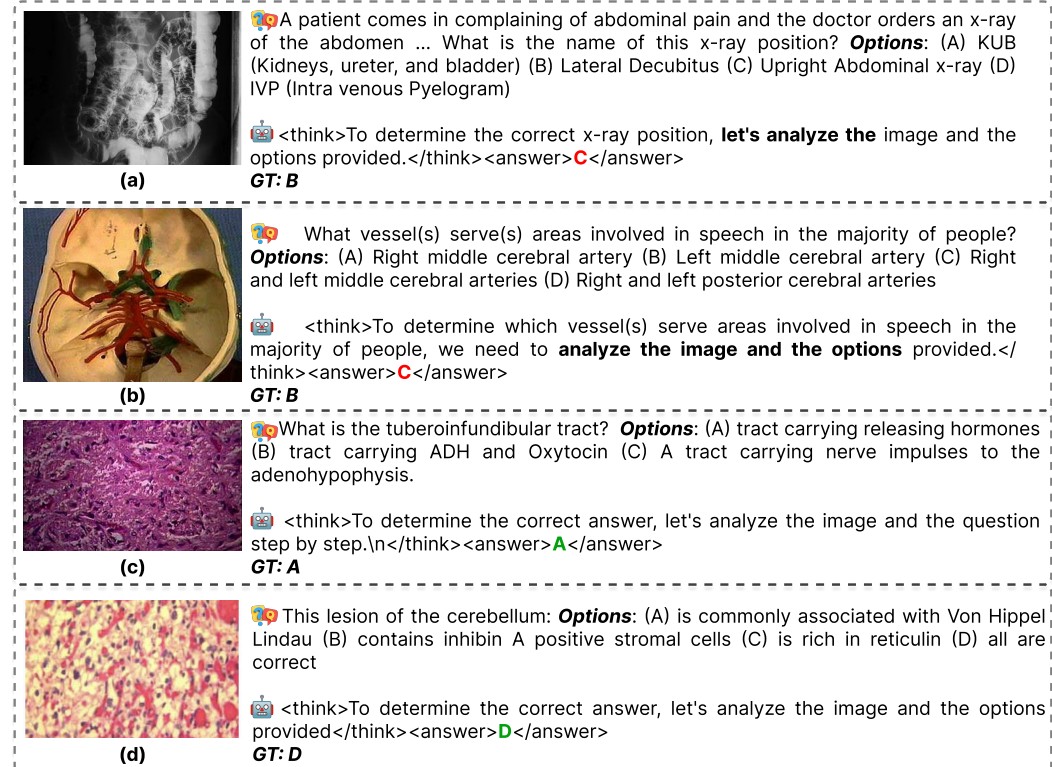

Figure 6: Failure case of MedVLM-R1 (Pan et al., 2025) on MMMU Medical Validation in (Chen et al., 2024b)

the individual pathological features that support the choice, thus yielding an incomplete reasoning chain.

In Figure 7, (a) in vestibular pathology, the model wrongly excludes loss of facial sensation, instead mislabeling vertigo as the least likely outcome, failing to distinguish cranial nerve VII vs VIII involvement. (b) For myocardial ischemia, the model selects heterophagocytosis instead of free radical injury, showing both a content error and neglect of pathophysiological time-course reasoning. (c) In cell classification, although the model outputs the correct choice ("eukaryotic with a nucleus"), the explanation is superficial and does not clearly contrast prokaryotic vs eukaryotic structures, reflecting step deficiency. (d) For regeneration of cranial nerves, the model outputs the correct "no regeneration" but again fails to elaborate the biological rationale (CNS vs PNS regenerative capacity), underscoring incomplete causal reasoning.

Taken together, these supplemental cases reinforce the observation that even when answers are sometimes correct, explanations are often *generic, incomplete, or clinically shallow*, leaving reasoning chains insufficient to support reliable medical decision-making.

## A.3 Step-wise Exploration & Branch-wise Exploration

### A.3.1 Exploration Details

In Section. 3.1, we employ LLM-as-Judge and prompt OpenAI GPT-4o (Achiam et al., 2023) to evaluate the step-wise and branch-wise exploration of each reasoning process. The detailed prompt template is provided in Appendix A.3.1. Based on the LLM-extracted reasoning steps for each candidate option, we compute $\mathrm{StepExploration}$ and $\mathrm{BranchExploration}$ according to Eq. 1. These two indicators quantitatively capture how progressively it develops causal or evidential links (step-wise exploration) and how comprehensively the reasoning process explores alternative diagnostic routes (branch-wise exploration). The corresponding statistics are reported in Table 1 and Table 2,

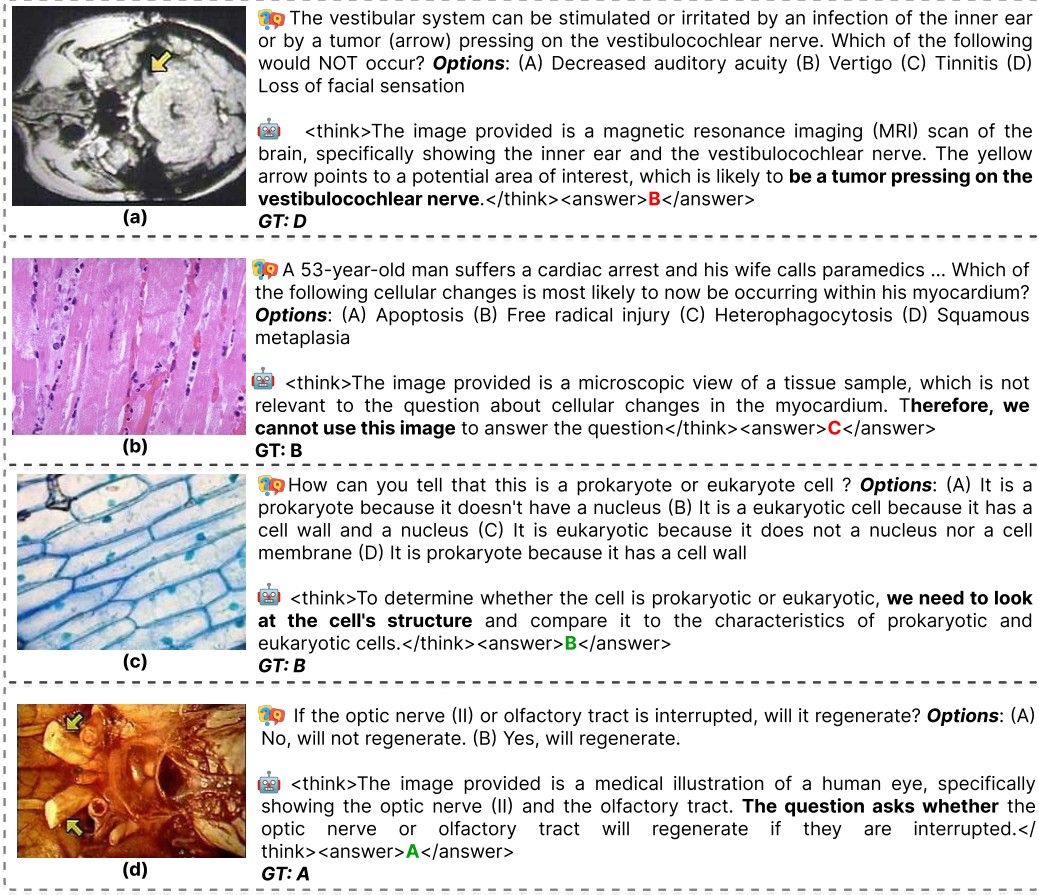

Figure 7: Failure case of Med-R1 (Lai et al., 2025) on MMMU Medical Validation in (Chen et al., 2024b)

providing complementary insights into the structural quality of reasoning beyond final-answer accuracy.

---

**Prompt Template**

You are given a medical multiple-choice question and an answer explanation generated by an AI model.
The **medical question** is: {QUESTION}.
The **answer explanation** is: {ANSWER_TEXT}.
Your Task:

- For each option, count the number of **distinct reasoning steps** in the explanation.
  (1) A reasoning step is a unique clinical statement or justification.
  (2) Do not count repeated or paraphrased content.
  (3) If an option is not mentioned, assign **0**.
- Report all options in order.

Output Format (Plain Text Only):
Option A: X
Option B: X
Option C: X
Option D: X

---

We illustrate the evaluation process with several representative cases. As shown in Figure 8, when the model produces a fake" reasoning path (e.g., generic statements such as let's analyze"), GPT-4o (Achiam et al., 2023) correctly judges that all options contain zero valid reasoning steps. In contrast, Figure 9 demonstrates how GPT-4o identifies the number of clinically meaningful reasoning steps for each option when valid explanations are present. To ensure alignment with human preference, we further conduct human verification by licensed physicians, confirming the reliability of GPT-4o as an automatic evaluator.

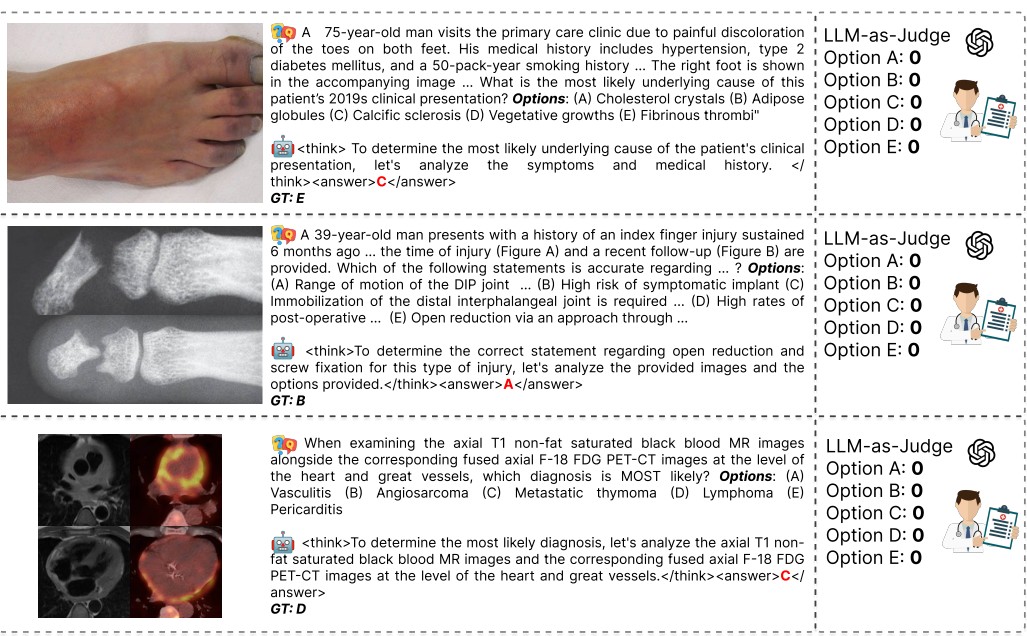

Figure 8: Evaluation provided by OpenAI GPT-4o (Achiam et al., 2023) to access the number of steps for each option in the reasoning process.

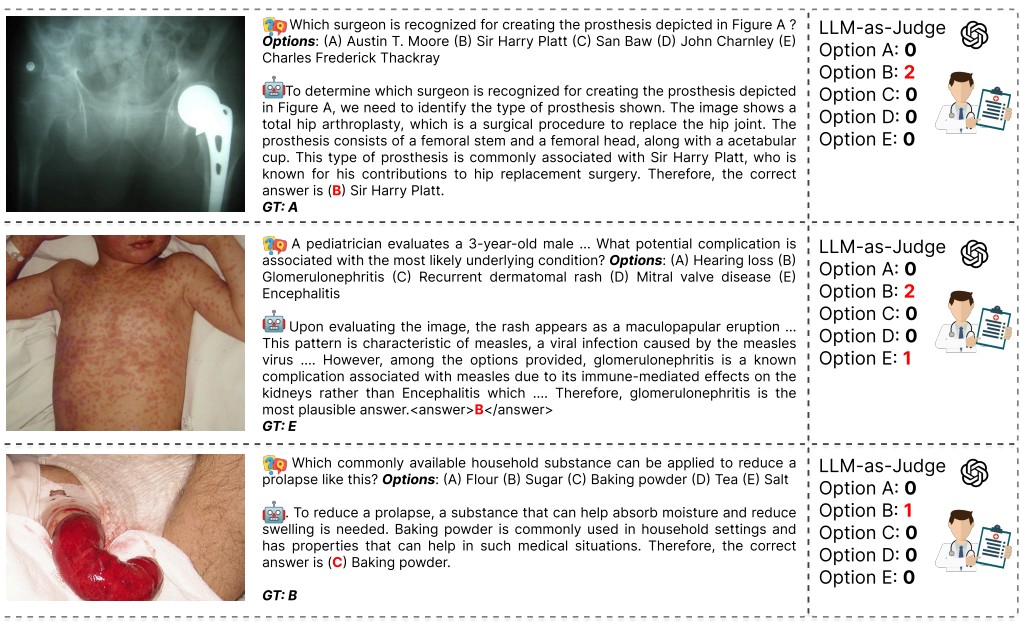

Figure 9: Evaluation provided by OpenAI GPT-4o (Achiam et al., 2023) to access the number of steps for each option in the reasoning process.

Table 6: **Inter-rater reliability between LLM-as-Judge scores and expert ratings** across four datasets for two reasoning-structure metrics. We report Spearman rank correlation ($\rho$). Higher $\rho$ indicates stronger agreement.

(a) Spearman correlation for BranchExploration.

| | MedX-M | MMMU | MedX-T | MMLU-Pro | **Avg.** |
|---|---|---|---|---|---|
| $\rho$-value | 0.849 | 0.884 | 0.824 | 0.826 | **0.846** |

(b) Spearman correlation for StepExploration.

| | MedX-M | MMMU | MedX-T | MMLU-Pro | **Avg.** |
|---|---|---|---|---|---|
| $\rho$-value | 0.818 | 0.858 | 0.813 | 0.801 | **0.822** |

### A.3.2 INTER-RATER RELIABILITY ANALYSIS

To validate the reliability of our LLM-as-Judge evaluation results in Table. 1, we first select 10 samples for each model (including 9 models: VL-Rethinker-7B, R1-Onevision-7B, R1-VL-7B, VLAAA-Thinker-7B, Med-R1-2B, MedVLM-R1-2B, Chiron-o1-8B, Lingshu-7B, and PathFiner-7B) on each dataset (including 4 datasets: MedX-M, MMMU-Med, MedX-T, MMLU-Pro), totally **360** samples and have them evaluated with same metrics by a licensed medical expert. Then we perform a quantitative inter-rater agreement analysis on these 360 samples, comparing LLM-as-Judge ratings with expert evaluation on Spearman correlation value. As shown in Table. 6, the Spearman $\rho$-value is **0.846** and **0.822**, demonstrating strong correlation between LLM-as-Judge and human expert ratings.

### A.4 METHODOLOGY

**Data Source for Graph-Structured Reasoning Representation.** For multimodal settings, our data source consists of PubMedVision (Chen et al., 2024b), MedTrinity-25M (Xie et al., 2025) and GMAI-Reasoning10k (Su et al., 2025). PubMedVision and MedTrinity-25M provide visual-question-answer pair accompanied with detailed image descriptions while GMAI-Reasoning10k provides visual-question-answer pair with tailed CoT reasoning process. We sample 10k, 10k and 9k VQA pairs from PubMedVision, MedTrinity-25M and GMAI-Reasoning10k datasets respectively. For text-only settings, our data source consists of QA pairs with detailed reasoning, including 32.7k from MedReason (Wu et al., 2025), 8k from medical-o1-reasoning-SFT (Chen et al., 2024a), 7k from Medical-R1-Distill-Data (Chen et al., 2024a), 23.5k from Medical23k (Huang et al., 2025a).

**Data Construction for Graph-Structured Reasoning Representation.** We construct our graph-based chain-of-thought (CoT) data by systematically integrating both multimodal and text-only medical sources. Our goal is to represent each diagnostic reasoning process as a structured graph, where nodes correspond to medical entities (e.g., symptoms, findings, hypotheses) and edges encode causal or evidential relationships. We illstrate the pipeline of constructing multimodal data as example: **(Step 1)**: Node Identification. For each question–answer pair, we first identify relevant medical entities to serve as graph nodes. To standardize this process, we prompt GPT-4o (Achiam et al., 2023) with a carefully designed template (Appendix A.4) to extract candidate entities from the textual content, and from image captions or clinical descriptions when available. Nodes are categorized into types such as findings, hypotheses, symptoms, or conclusions, ensuring consistent semantic representation across datasets. **(Step 2)**: Edge Construction and Reasoning Paths. Once the nodes are defined, we generate edges that capture causal or evidential relationships between nodes. For each answer option, GPT-4o is prompted (Appendix A.4) to produce reasoning paths connecting the nodes in a coherent, step-wise manner. Each edge is annotated to indicate the type of relationship to facilitate downstream process-level reward modeling in Graph-GRPO. **(Step 3)**: Graph Reformatting. The generated reasoning paths are then reorganized into a graph-based CoT format, where multiple paths corresponding to different answer options are represented in a consistent structure. This enables explicit step-wise and branch-wise exploration during model training. **(Step 4)**: Quality Verification. To ensure clinical reliability, all generated graphs undergo review by licensed medical experts. Experts verify node correctness, edge validity, and the logical consistency of reasoning paths. Any inconsistencies or missing links are corrected before finalizing the CoT data. Finally, we analyze the frequency of nodes and edges across all datasets to identify the most common medical entities and reasoning relationships, which are visualized in Figure 10. This analysis provides an overview of the typical structure and distribution of graph-structured reasoning in medical VLM tasks.

**Dataset-Specific Adaptations.** Multimodal datasets (PubMedVision, MedTrinity-25M): We utilize both the question–answer pairs and the accompanying detailed image descriptions or captions, enabling GPT-4o (Achiam et al., 2023) to generate nodes and reasoning paths that integrate visual and textual evidence. Multimodal datasets (GMAI-Reasoning10k): As this dataset provides detailed reasoning without descriptive context, we modify the prompting procedure to omit description-based inputs while still constructing complete reasoning graphs. Text-only datasets: For datasets without images, GPT-4o is prompted using the original reasoning traces as references, ensuring that nodes and edges remain consistent with the underlying medical logic.

---

**Prompt Template**

You are an AI assistant specialized in biomedical and medical topics. You will receive a **medical question**, and/or **description of the image** and/or **reference reasoning** and the **image**. Your task is to **only extract related clinical entities or phrases from the description** and categorize each into one of the following types: $< symptom >$, $< finding >$, $< hypothesis >$, $< fact >$, $< rule >$, $< evidence >$, $< imag\_feature >$, $< roi >$, $< question >$, and other important types et al.
**Output Format (strictly follow this template):**
name of entity 1: $< type >$
name of entity 2: $< type >$
... ...
The **medical question** is: {QUESTION}.
The **description of the image** is: {DESCRIPTION}.
The **reference reasoning** is: {REASONING}.
The **image** is: {Encoded_Image}.

---

---

**Prompt Template**

You are an AI assistant specialized in biomedical and medical topics. You will receive a **medical question**, a **description** of an image, **incorrect answer**, **correct answer, and reference **reason**. Your task is to **construct reasoning chains that rule out the incorrect answer** and **construct reasoning chains that support the incorrect answer** using **important entities** and logic from the caption or question. You should only keep important medical entities $< tag >$ that are related to the question and options. You should include the type of important entity inside the $<>$, following the entity name. The reasoning should contain important steps that clearly explain why the incorrect answer is incorrect and why the correct answer is correct.

**Output Format Example:**

... entity $< type >$ ... entity $< type >$ ... $< refute >$ $< option >$ X.

... entity $< type >$ ... entity $< type >$ ... $< support >$ $< option >$ X.

Strictly Follow Output Format Requirements:

1. Replace the $< type >$ with the type of the entity, such as $< symptom >$, $< finding >$, $< hypothesis >$, $< fact >$, $< rule >$, $< evidence >$ and et al.

2. Replace the $entity$ with the actual entity name.

3. The $entity$ and $< type >$ should come from the **question entity and its type** and the **image entity and its type**.

4. You can replace the $< support >$ and $< refute >$ with any other tag that you think is more appropriate, such as $< support >$, $< justify >$, $< rule\ out >$, $< lead\ to >$ and et al.

5. Replace all the ... with the actual reasoning process complete sentence and make the reasoning process as detailed.

6. The reasoning should be based on the **caption**, the reference **reason** and the **image**.

7. Fully consider the underlying medical knowledge and logic to explain.

The **medical question** is: {QUESTION}.

The **description of the image** is: {DESCRIPTION}.

The **reference reasoning** is: {REASONING}.

The **image** is: {Encoded_Image}.

The **entity nodes** are: {NODES}

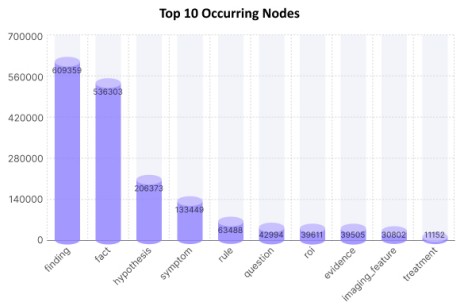

(a) Top 10 occurring nodes.

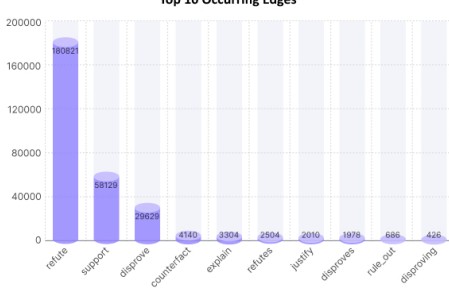

(b) Top 10 occurring edges.

Figure 10: Statistics of top 10 occurring nodes and edges in the structured graph.

**Data Source for Graph-based Group Relative Policy Optimization.** We acute both multimodal and text-only data for reinforcement learning stage. Specifically, we sample 1.6k from PMC-VQA (Zhang et al., 2023), 1.6k from PathVQA (He et al., 2020), 1.6k form SLAKE (Liu et al., 2021), 1k from VQA-RAD (Lau et al., 2018) and 1.2k from GMAI-Reasoning10K (Su et al., 2025) as multimodal dataset and 7k from MedAgentsBench (Tang et al., 2025), including training samples from MMLU (Hendrycks et al., 2020), PubMedQA (Jin et al., 2019), MedMCQA (Pal et al., 2022), MedQA (Jin et al., 2021) and Medbullets (Chen et al., 2025a) as text-only dataset.

Table 7: Comprehensive evaluation on medical report generation tasks.

| Model | CheXpert Plus | | | IU-Xray | | |
|---|---|---|---|---|---|---|
| | ROUGE-L | CIDEr | RaTE | ROUGE-L | CIDEr | RaTE |
| *Close-source proprietary models* | | | | | | |
| GPT-4.1 | 24.5 | 78.8 | 45.5 | 30.2 | 124.6 | 51.3 |
| Claud Sonnet 4 | 22.0 | 59.5 | 43.5 | 25.4 | 88.3 | 55.4 |
| Gemmi-2.5-Flash | 23.6 | 72.2 | 44.3 | 33.5 | 129.3 | 55.6 |
| *Open-source models (<10B)* | | | | | | |
| Med-R1-2B | 18.6 | 37.1 | 38.5 | 16.1 | 38.3 | 41.4 |
| MedVLM-R1-2B | 20.9 | 43.5 | 38.9 | 22.7 | 61.1 | 46.1 |
| MedGemma-4B-IT | **27.1** | 79.0 | **47.2** | 30.8 | 103.6 | 57.0 |
| LLaVA-Med-7B | 18.4 | 45.5 | 38.8 | 18.8 | 68.2 | 40.9 |
| HuatuoGPT-V-7B | 21.3 | 64.7 | 44.2 | 29.6 | 104.3 | 52.9 |
| BioMediX2-8B | 18.1 | 47.9 | 40.8 | 19.6 | 58.8 | 40.1 |
| Qwen2.5VL-7B | 22.2 | 62.0 | 41.0 | 26.5 | 78.1 | 48.4 |
| InternVL2.5-8B | 20.6 | 58.5 | 43.1 | 24.8 | 75.4 | 51.1 |
| InternVL3-8B | 20.9 | 65.4 | 44.3 | 22.9 | 76.2 | 51.2 |
| Lingshu-7B | 26.5 | 79.0 | 45.4 | **41.2** | **180.7** | 57.6 |
| *PathFinder-7B* (Ours) | 26.1 | **88.7** | 46.9 | 35.0 | 153.6 | **63.5** |

## A.5 EXPERIMENTAL SETUPS

**Training Details.** For cold-start SFT, we utilize LlamaFactory (Zheng et al., 2024) as codebase. We fully finetune language model of Qwen2.5-VL-7B-Instruct (Bai et al., 2025) by freezing the vision encoder and multimodal projector, for 2 epoch with learning rate $2e^{-5}$, cosine scheduler and deepspeed zero3 stage. For reinforcement learning, we apply our proposed Graph-GRPO on VLM-R1 (Shen et al., 2025) codebase. Due to 4×A6000-48GB GPU memory limitation, we utilize LoRA (Hu et al., 2022) with 64 rank, 128 alpha and 0.05 dropout, with learning rate $1e^{-5}$ and 4 number of generations to train 1 epoch. The rest of parameter settings follow the default of LlamaFactory and VLM-R1 codebases.

## A.6 EXPERIMENTAL RESULTS

**Close-End Task** In Table. 1, VL-Rethinker-7B (Wang et al., 2025), R1-Onevision-7B (Yang et al., 2025), R1-VL-7B (Zhang et al., 2025) and VLAA-Thinker-7B (Chen et al., 2025b) are utilized as general models. Med-R1-2B (Lai et al., 2025), MedVLM-R1-2B (Pan et al., 2025), Chiron-8B (Sun et al., 2025a) and Lingshu-7B (Xu et al., 2025b) represent the medical domain-specific models. All models are inferenced following their official settings to produce the reasonings. Thus, the results in accuracy slightly differ from the leaderboards in Table. 3 and Table. 4. In Table. 3, we follow the leaderboards in (Xu et al., 2025b) which includes GPT-4.1, Claude Sonnet and Gemini-2.5-Flash as close-source proprietary models, BiomedGPT (Zhang et al., 2024), MedGemma-4B (Sellergren et al., 2025), LLaVA-Med (Li et al., 2023), HuatuoGPT-V-7B (Chen et al., 2024b) and BioMediX2-8B (Mullappilly et al., 2024) as open-source non-reasoning models, as well as Med-R1-2B (Lai et al., 2025), MedVLM-R1-2B (Pan et al., 2025), Lingshu-7B (Xu et al., 2025b), Chiron-o1-8B (Sun et al., 2025a) and MedVLThinker-7B (Huang et al., 2025a) as open-source reasoning-based models. Since the orignal leaderboards in (Xu et al., 2025b) do not include Chiron-o1-8B and MedVLThinker-7B, we add these two results from their own papers, where MedVLThinker-7B does not report results on OmniMedVQA dataset and any text-only benchmarks. Even we sample data from GMAI-Reasoning10K (Su et al., 2025), we exclude comparison of GMAI-VL-R1 (Su et al., 2025) since the model has yet been released at the time of our paper submission and (Su et al., 2025) its paper only reports on a small number of benchmark.

**Open-End Task** In addition to conventional QA and VQA benchmarks, we further evaluate the performance of our PathFinder model on a practical task of high clinical relevance: medical report generation, a setting where there are no predefined candidate answers. We adopt the same

SFT→Graph-GRPO training pipeline used for closed-ended tasks, with adaptations to graph construction and reward design. We randomly sample 4k data (2k for SFT and 2k for RL) from the training split of CheXpert Plus dataset (Chambon et al., 2024). (1) **Graph Construction:** For each training sample, we constructs a diagnostic graph using the same node and edge schema as in Section 3. Nodes are extracted and edges are generated from free-form relations among these entities, capturing both confirmatory and eliminative reasoning even without explicit answer options. (2) **SFT Data Generation:** The diagnostic graph is unfolded into a reasoning trajectory that precedes the final report. The output format follows the template: ... *'Reasoning Process' ... <Final Report> Findings: . . . Impression: . . . < /Final Report>*. Although the task is open-ended, the reasoning path still includes exploration of alternative hypotheses and rule-out logic, enabling PathFinder to learn structured reasoning similar to differential diagnosis. (3) **Graph-GRPO Adaptation for Open-Ended Output:** Process-level rewards (*Step Reward* and *Branch Reward*) remain unchanged: Step Reward verifies that each reasoning step correctly uses nodes and edges from the constructed graph, while Branch Reward encourages exploration of multiple inferred hypotheses. Since open-ended tasks do not provide categorical correctness, the outcome reward is replaced with a sentence-level semantic reward. Specifically we adopt BERTScore F1, continuous value between 0.0 and 1.0 following (Pham & Ngo, 2025). The final reward is $r = \lambda_{\text{step}} r_{\text{step}} + \lambda_{\text{branch}} r_{\text{branch}} + \lambda_{\text{acc}} \text{BERTScore\_F1}(y, \hat{y})$. These hyperparameters $\lambda_{\text{step}}, r_{\text{step}}, \lambda_{\text{branch}}, r_{\text{branch}}, \lambda_{\text{acc}}$ are consistent with close-end task defined in Section. 3.3. The results are summarized in Table 7, including two widely adopted benchmarks: CheXpert Plus (Chambon et al., 2024), and IU-Xray (Demner-Fushman et al., 2015). We follow the leaderboard in (Xu et al., 2025b) and specifically report two semantic-based metrics ROUGE-L and CIDEr, one model-based metrics RaTE. Our PathFinder-7B model achieves competitive or superior results compared with both open-source and proprietary models. On CheXpert Plus, PathFinder obtains a CIDEr of 88.7, surpassing all open-source baselines and proprietary model performance. Notably, it achieves the highest RaTE score (63.5) on IU-Xray, indicating strong clinical correctness and relevance in generated reports. These results demonstrate that PathFinder can generate detailed and clinically accurate reports, confirming its effectiveness in open-end medical language tasks beyond traditional (V)QA benchmarks. We provide a visualization example in Figure 11.

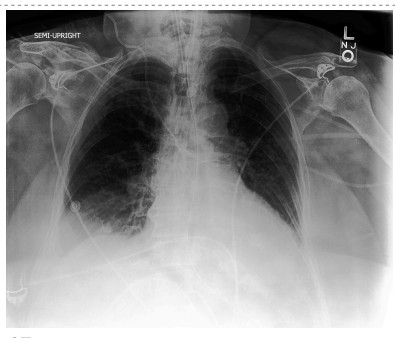

SEMI-UPRIGHT

**Ours-7B**

Right internal jugular central venous catheter <finding> is visualized coursing from the neck into the thorax with its tip projecting over the expected region of the superior vena cava on this frontal semi-upright radiograph, based on the image <support> this supports appropriate intravascular placement of the line. No pneumothorax <finding> is reported and, based on the image, there is no visible pleural line or focal increased radiolucency at the right lung apex or along the right hemithorax to suggest free intrapleural air, therefore <refute> pneumothorax. Right lower lobe consolidation <imaging_feature> appears as a new dense opacity at the right base on this film, and given the patient's known multiple myeloma <fact>, this finding is concerning for possible infection or myeloma-related consolidation, so <support> clinical correlation for symptoms of pneumonia. Small bilateral pleural effusions <imaging_feature> are suggested by subtle dependent layering and blunting at the bases on this frontal view, and the presence of external monitoring leads/wires <finding> does not obscure these basilar changes, supporting the impression of small effusions <support>. Cardiomediastinal silhouette stable <finding> indicates no acute cardiomegaly or large-volume mediastinal shift despite the new right basilar opacity, which further <support> that the abnormality is a localized right lower lobe process rather than a diffuse cardiogenic or mediastinal cause.
**<Final Report>**: Findings: Right internal jugular central venous catheter in expected intravascular position; no pneumothorax identified; **right lower** lobe consolidation; **small bilateral pleural effusions**; cardiomediastinal silhouette stable; external monitoring leads/wires present. Impression: 1) Right internal jugular central venous catheter appropriately positioned with no radiographic evidence of pneumothorax. 2) New right lower lobe consolidation concerning for infection versus myeloma-related consolidation - correlate clinically. 3) Small bilateral pleural effusions. **</Final Report>**

*GT:*

*Findings:* Portable chest shows low lung volumes with crowding of the pulmonary vasculature. The lines and tubes are stable, except the endotracheal tube has been pulled back to 7.9 cm above the carina. There is bilateral **lower** lobe airspace disease with partial clearing of **the right lung** base. This is the suggestion of **small pleural fluid** collections. Otherwise, there is no change from the prior examination.

*Impression:* 1.ENDOTRACHEAL TUBE IN HIGH POSITION ABOVE CARINA.

Figure 11: Example of report generation on CheXpert Plus (Chambon et al., 2024) dataset.

**Training Process** We visualize the training progress of Graph-GRPO by plotting the Step Reward ($r_{\text{step}}$), Branch Reward ($r_{\text{branch}}$) and Accuracy Reward ($r_{\text{acc}}$) over the first 50 global steps. As shown in Figure 12, both $r_{\text{step}}$ and $r_{\text{branch}}$ increase steadily alongside the accuracy reward, indicating

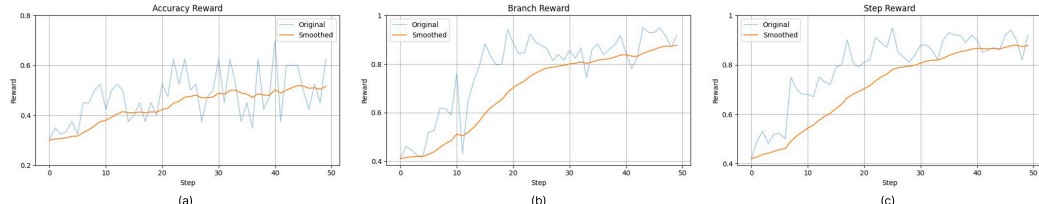

Figure 12: Training progress over the first 50 global steps. We visualize the Accuracy Reward, Branch Reward, and Step Reward, showing both the original data and the time-weighted EMA-smoothed curves.

that the model learns to improve step-wise and branch-wise reasoning in a balanced manner. This demonstrates that these rewards can be jointly optimized without conflict during the training stage.

## A.7 ABLATION STUDY

We conduct full ablation studies on Qwen2.5-VL-7B across both text-only and multimodal benchmarks, as shown in Tables 8 and 9. Starting from the baseline model, we first add a *Cold-Start* stage, which consistently improves performance across both in-domain and out-of-domain datasets. Building on this, integrating our proposed **Graph-GRPO** further enhances accuracy, yielding the highest gains in both text-only (Table 9) and multimodal (Table 8) settings. These results confirm that (i) Cold-Start training provides a stronger initialization and (ii) Graph-GRPO offers additional improvements by explicitly optimizing step- and branch-wise reasoning quality.

Table 8: Full ablation study on Qwen2.5-VL-7B across multimodal benchmarks.

| Models | Out-of-Domain | | In-Domain | | | | |
| | MedXQA↑ | MMMU-Med↑ | VQA-RAD↑ | SLAKE↑ | PathVQA↑ | PMC-VQA↑ | OMVQA↑ |
| --- | --- | --- | --- | --- | --- | --- | --- |
| Baseline-7B | 22.2 | 50.6 | 64.5 | 67.2 | 44.1 | 51.9 | 58.4 |
| + Cold-start | 26.1 | 62.8 | 77.1 | 77.2 | 75.2 | 56.6 | 60.4 |
| + Graph-GRPO | 28.2 | 68.9 | 79.3 | 80.3 | 87.0 | 61.2 | 63.5 |

Table 9: Full ablation study on Qwen2.5-VL-7B across text-only benchmarks.

| Models | Out-of-Domain | | In-Domain | | | | | |
| | MedXQA↑ | SGPQA-Med↑ | MMLU↑ | PubMedQA↑ | MedMCQA↑ | MedQA↑ | Medbullets↑ | MMLU-Pro↑ |
| --- | --- | --- | --- | --- | --- | --- | --- | --- |
| Baseline-7B | 12.9 | 26.3 | 73.4 | 73.1 | 52.6 | 57.3 | 42.1 | 46.5 |
| + Cold-Start | 15.0 | 30.5 | 78.1 | 75.4 | 54.3 | 64.1 | 43.1 | 54.4 |
| + Graph-GRPO | 16.8 | 31.5 | 80.2 | 77.2 | 56.8 | 66.3 | 46.4 | 55.4 |

## A.8 SENSITIVITY ANALYSIS

To evaluate the robustness of PathFinder with respect to hyperparameter choices, we conducted a sensitivity analysis by varying the temperature $\tau$ in the model's output sampling. Table 10 reports the performance on the MedXpert-MM (Zuo et al., 2025) benchmark under different temperature settings ($\tau = 1.0, 0.75, 0.50, 0.25$). Typically, we report $\tau = 1$ in all other table results through this paper. We observe that both PathFinder-SFT and PathFinder-RL maintain stable performance across a wide range of $\tau$, with small standard deviations, indicating that the model's reasoning quality and final prediction are robust to stochasticity in sampling. In particular, PathFinder-RL consistently outperforms the baseline Qwen2.5VL-7B under all tested temperatures, demonstrating the effectiveness of graph-structured reasoning and the RL-based fine-tuning in producing reliable answers.

Table 10: Sensitivity analysis by setting different temperature $\tau$ hyperparameter.

| Models | MedX-M | | | | |
| | $\tau = 1$ | $\tau = 0.75$ | $\tau = 0.50$ | $\tau = 0.25$ | Average |
|---|---|---|---|---|---|
| Baseline-7B | 22.2 | 22.0 | 21.0 | 21.1 | 21.6± 0.53 |
| + Cold-start | 26.1 | 26.5 | 26.0 | 26.5 | 26.3±0.23 |
| + Graph-GRPO | 28.2 | 28.4 | 26.5 | 28.5 | 27.9±0.82 |

## A.9   OURS CASE STUDY

We provide both accurate and failed cases of our model compared with Med-R1 (Lai et al., 2025), MedVLM-R1 (Pan et al., 2025), Chiron-o1 (Sun et al., 2025a) and Lingshu (Xu et al., 2025b) on MedXpertQA-MM benchmark (Zuo et al., 2025).

In Figure 13, a 75-year-old man with hypertension, diabetes, and a heavy smoking history presented with painful toe discoloration and a pulsatile abdominal mass, with intact peripheral pulses. The ground-truth diagnosis is fibrinous thrombi due to embolization from an abdominal aortic aneurysm. MedVLM-R1 misclassifies the case as calcific sclerosis, reflecting a superficial reliance on atherosclerosis-related cues without integrating the acute ischemic manifestation or the abdominal finding. In contrast, our model correctly identify fibrinous thrombi by explicitly linking the ischemic discoloration to distal embolization and connecting the pulsatile abdominal mass to an underlying aneurysm, thereby capturing both step-wise exploration (reasoning through the causal chain from aneurysm to thromboembolism) and branch-wise exploration (considering risk factors and multiple findings). This illustrates how our approach mitigates distractor bias and achieves more faithful pathophysiological reasoning.

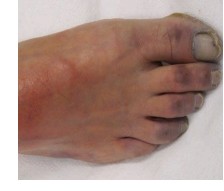
🔺❓A 75-year-old man visits the primary care clinic due to painful discoloration of the toes on both feet. His medical history includes hypertension, type 2 diabetes mellitus, and a 50-pack-year smoking history, with ongoing smoking of one pack per day. He denies illicit drug use or recent contact with sick individuals. His temperature is 98.6\u00b0F, blood pressure is 140/90 mmHg, pulse is 84/min, and respiratory rate is 12/min. On examination, a non-tender pulsatile mass is palpated along the abdominal midline. The right foot is shown in the accompanying image. Popliteal and pedal pulses are +2 bilaterally. What is the most likely underlying cause of this patient\u2019s clinical presentation? **Options**: (A) Cholesterol crystals (B) Adipose globules (C) Calcific sclerosis (D) Vegetative growths (E) Fibrinous thrombi
***GT: E***

🤖 MedVLM-R1-2B

\<think\> To determine the most likely underlying cause of the patient's clinical presentation, let's analyze the symptoms and medical history. \</think\>\<answer\>**C**\</answer\>

🤖 Ours-7B

Based on the image showing \<finding\> painful discoloration of the toes \</finding\>, it indicates significant vascular compromise consistent with \<hypothesis\> thromboembolic events \</hypothesis\>. The presence of a \<finding\> pulsatile mass \</finding\> in the abdominal midline suggests a potential \<hypothesis\> abdominal aortic aneurysm (AAA) \</hypothesis\>, which is known to lead to complications such as \<fact\> emboli \</fact\> that travel to the lower extremities, causing ischemia and resulting in the observed discoloration. Therefore, the most likely underlying cause of this patient's clinical presentation is \<support\> option E: Fibrinous thrombi \</support\>.\n\nIn contrast, the option A: Cholesterol crystals does not align with the clinical findings. The presence of a \<finding\> pulsatile mass \</finding\> indicates a vascular issue rather than a metabolic one associated with cholesterol crystals, which typically do not cause such symptoms. Additionally, the patient's history of \<fact\> ongoing smoking \</fact\> and \<fact\> type 2 diabetes mellitus \</fact\> further supports the likelihood of thromboembolic events over metabolic causes, thus \<refute\> option A: Cholesterol crystals \</refute\>.\n\nSimilarly, for option B: Adipose globules, the clinical presentation of \<finding\> painful discoloration of the toes \</finding\> suggests ischemic changes rather than a condition related to adipose tissue. The pulsatile mass in the abdomen is more indicative of an \<hypothesis\> abdominal aortic aneurysm (AAA) \</hypothesis\>, which can lead to emboli affecting peripheral circulation, making \<refute\> option B: Adipose globules \</refute\> incorrect.\n\nRegarding option C: Calcific sclerosis, the patient's symptoms of \<symptom\> painful discoloration of the toes \</symptom\> indicate ischemia, which is not typically associated with calcific sclerosis. This condition usually presents with chronic pain and does not correlate with the acute ischemic changes seen here. Thus, \<refute\> option C: Calcific sclerosis \</refute\> is also incorrect.\n\nLastly, option D: Vegetative growths is ruled out because the clinical signs suggest vascular issues rather than infectious processes. The presence of a \<finding\> pulsatile mass \</finding\> and the patient's risk factors point towards an \<hypothesis\> abdominal aortic aneurysm (AAA) \</hypothesis\> leading to emboli, making \<refute\> option D: Vegetative growths \</refute\> incorrect.\n\<answer\>**E**\</answer\>

Figure 13: Comparison between MedVLM-R1 (Pan et al., 2025) and Ours.

In Figure 14, when evaluating axial T1 non-fat saturated black blood MR images with corresponding fused FDG PET-CT at the level of the heart and great vessels, the ground-truth diagnosis was lymphoma, characterized by high metabolic uptake in the mediastinal region. Med-R1 incorrectly predicts metastatic thymoma, likely due to overfitting to the presence of mediastinal involvement

without recognizing the diffuse and metabolically aggressive pattern. In contrast, our model correctly identify lymphoma by integrating the imaging findings of markedly increased metabolic activity with the anatomical context of mediastinal lymph node involvement, while systematically excluding vasculitis, angiosarcoma, metastatic thymoma, and pericarditis. This case highlights how our method leverages structured step (reasoning through pathophysiological hallmarks such as metabolic aggressiveness) and branch (considering all plausible differential diagnoses) to reach the correct conclusion where baseline models fail.

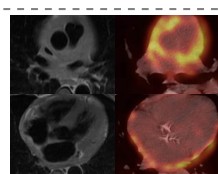

🧑‍🔬 When examining the axial T1 non-fat saturated black blood MR images alongside the corresponding fused axial F-18 FDG PET-CT images at the level of the heart and great vessels, which diagnosis is MOST likely? **Options**: (A) Vasculitis (B) Angiosarcoma (C) Metastatic thymoma (D) Lymphoma (E) Pericarditis
*GT: D*

🤖 Med-R1-2B

<think>To determine the most likely diagnosis, **let's analyze the axial** T1 non-fat saturated black blood MR images and the corresponding fused axial F-18 FDG PET-CT images at the level of the heart.</think><answer>C</answer>

🤖 Ours-7B

Based on the image showing the axial T1 non-fat saturated black blood MR images and the corresponding fused axial F-18 FDG PET-CT images, the findings indicate significant <finding> increased metabolic activity </finding> in the region of the <roi> mediastinum </roi>. This high metabolic activity is characteristic of <hypothesis> lymphoma </hypothesis>, as it often presents with such features due to its aggressive nature and tendency to involve the mediastinal lymph nodes. Therefore, this supports option <option>D</option>.\n\nIn contrast, for option <option>A</option> (Vasculitis), the imaging does not show any signs of inflammation or vascular abnormalities typically associated with vasculitis. The lack of these features, along with the presence of increased metabolic activity, <refutes> option A.\n\nFor option <option>B</option> (Angiosarcoma), while angiosarcoma can present with increased metabolic activity, the imaging characteristics do not align with those expected from angiosarcoma, which usually shows distinct vascular structures. The absence of these features further <refutes> option B.\n\nRegarding option <option>C</option> (Metastatic thymoma), although metastatic thymoma may exhibit increased metabolic activity, the imaging findings suggest a more diffuse pattern rather than the localized involvement typical of metastatic thymoma. Thus, this discrepancy <refutes> option C.\n\nLastly, for option <option>E</option> (Pericarditis), the imaging does not reveal any signs of inflammation or fluid accumulation around the heart, which are key indicators of pericarditis. The observed increased metabolic activity is inconsistent with pericarditis, thereby <refuting> option E.\n<answer>D</answer>

Figure 14: Comparison between Med-R1 (Lai et al., 2025) and Ours.

In Figure 15 trauma case, a 32-year-old woman with bilateral anterior sacroiliac joint diastasis underwent pelvic stabilization with iliac and sacroiliac screws. The ground-truth complication is dyspareunia, a common long-term sequela resulting from pelvic structural changes after such surgery. Chiron-o1 incorrectly predicts symptomatic anterior pelvis instability, likely focusing only on mechanical stability while overlooking functional complications specific to pelvic anatomy. In contrast, our model correctly identifies dyspareunia by integrating surgical context with clinical sequelae, reasoning that the primary risk is not failure of fixation but altered pelvic biomechanics affecting intercourse. By systematically excluding unrelated options such as ankle weakness, hardware symptoms, and thigh adduction weakness, our method demonstrated superior step-wise exploration and branch-wise exploration of reasoning, aligning with the ground-truth outcome.

In Figure 16, a 12-year-old girl presented with a prominent tibial mass below the knee that was asymptomatic at rest but tender on impact, with imaging and histology provided. The ground-truth diagnosis is osteofibrous dysplasia. The Lingshu model incorrectly predicts an aneurysmal bone cyst, over-interpreting the lytic and septated appearance as cystic rather than fibro-osseous. In contrast, our model correctly identifies osteofibrous dysplasia by integrating clinical presentation (absence of systemic symptoms and preserved function), imaging features (soap-bubble-like lytic lesion with benign margins), and histologic evidence of fibrous tissue within bone trabeculae. Moreover, it systematically ruled out other differentials: low-grade chondrosarcoma and adamantinoma due to their aggressive course, aneurysmal bone cyst due to greater symptomatic burden, and non-ossifying fibroma due to mismatch in clinical prominence. This case highlights our model's ability to align multimodal evidence into a coherent diagnostic reasoning chain, distinguishing benign fibro-osseous pathology from cystic and malignant mimics.

For comparison, we include cases where our model fails to provide the correct answers while Chiron-o1 and Lingshu predict correctly. In Figure 17, our method misclassifies the erythrocyte inclusions as basophilic stippling corresponding to ribosomal RNA accumulation, leading to the incorrect selection of B. Ribosomes. The error arose from over-interpreting the Giemsa smear features while

🔬🗨 A 32-year-old woman arrives at the trauma bay following a head-on motor vehicle collision. She has sustained multiple facial fractures and the injury depicted in Figure A. A CT scan confirms bilateral anterior sacroiliac (SI) joint diastasis. After successful resuscitation, she undergoes the procedure shown in Figure B. Which long-term complication is this patient most at risk of developing? *Options*: (A) Ankle plantar flexion weakness (B) Symptomatic hardware (C) Symptomatic anterior pelvis instability (D) Thigh adduction weakness (E) Dyspareunia
*GT: E*

🖥 Chiron-o1-8B

Step 1: \nEvaluate the mechanism of injury and the surgical intervention. The patient sustained a high-energy trauma with bilateral anterior sacroiliac joint diastasis, which is a significant injury to the pelvic ring. The surgical intervention shown in Figure B involves the placement of bilateral iliac screws and a sacroiliac screw, which is a standard procedure to stabilize the pelvic ring and restore anterior pelvic stability.\n\n### Step 2:\nConsider the potential long-term complications associated with this type of surgical intervention. The primary concern with such stabilization is maintaining pelvic stability while minimizing complications. The most common long-term complications include hardware-related issues (such as pain or hardware failure) and instability-related issues (such as chronic pelvic pain or dysfunction). \n\n### Step 3:\nAnalyze the specific risks associated with the hardware and the surgical technique used. The bilateral iliac screws and sacroiliac screw are designed to provide robust stabilization. However, the hardware may sometimes cause symptoms such as pain or discomfort, especially if it leads to altered biomechanics or nerve irritation. While hardware-related symptoms are possible, they are not the most common complication in this context.\n\n### Step 4:\nAssess the risk of anterior pelvic instability. Given the nature of the injury (bilateral anterior sacroiliac joint diastasis) and the surgical goal (restoration of anterior pelvic stability), the patient is at significant risk for developing symptomatic anterior pelvic instability if the stabilization is not successful. This can manifest as chronic pelvic pain, instability, or functional limitations, which are well-documented long-term complications of such injuries and surgeries.\n\n### Step 5:\nCompare the risks with other potential complications. While other complications like ankle plantar flexion weakness, thigh adduction weakness, or dyspareunia are possible, they are less directly related to the specific surgical intervention and the primary injury pattern. These complications are more likely to arise from other factors or secondary injuries sustained in the trauma.\n\n### The final answer is: **C**. Symptomatic anterior pelvis instability

🖥 Ours-7B

Based on the image showing the surgical intervention involving <screws and plates> <finding> for the <bilateral anterior sacroiliac (SI) joint diastasis> <finding>, it is evident that the patient underwent a <surgical stabilization> <fact> to address her pelvic injuries. This surgical intervention carries significant risks, particularly regarding <symptom> dyspareunia, which is characterized by pain during intercourse due to <pelvic structural changes> <hypothesis> resulting from the surgery. Therefore, option E, <dyspareunia> <symptom>, is supported as the most likely long-term complication.\n\nIn contrast, option A, <ankle plantar flexion weakness> <symptom>, is not relevant because the surgical focus was on the pelvic region, specifically the <sacroiliac joints> <roi>. The complications associated with this type of surgery do not typically involve ankle function, thus <refuting> option A. \n\nSimilarly, option B, <symptomatic hardware> <symptom>, while a potential concern after surgery, does not represent the primary long-term complication expected from the surgical stabilization of the <sacroiliac joints> <roi>. The more pressing issue is the risk of <symptom> dyspareunia due to <pelvic structural changes> <hypothesis>, making option B less likely and <disproving> its relevance.\n\nFor option C, <symptomatic anterior pelvis instability> <symptom>, although it may seem plausible given the surgical intervention, the focus of the surgery was to stabilize the <sacroiliac joints> <roi> rather than directly addressing anterior pelvic instability. Thus, the risk of instability is not the primary concern, leading to the <counterfact> of option C being incorrect.\n\nLastly, option D, <thigh adduction weakness> <symptom>, is also unlikely since the surgical intervention primarily involved the <sacroiliac joints> <roi> and not the muscles responsible for thigh adduction. Therefore, this option can be <disproved> as well.\n\nIn summary, the reasoning clearly supports option E, <dyspareunia> <symptom>, as the most likely long-term complication arising from the surgical stabilization of the <sacroiliac joints> <roi>, while effectively ruling out options A, B, C, and D based on their lack of relevance to the surgical context.\n<answer>E</answer>

Figure 15: Comparison between Chiron-o1 (Sun et al., 2025a) and Ours.

overlooking the confirmatory Prussian blue staining, which specifically highlights iron deposits. As a result, our approach fails to integrate cross-modal evidence from both stains and instead focused narrowly on a single morphological pattern. By contrast, the correct diagnosis A. Iron is supported by the Prussian blue positivity, which directly indicates sideroblastic changes with iron accumulation. This case illustrates a failure mode where the model over-relies on surface morphological resemblance without grounding the reasoning in stain-specific biochemical validation.

In Figure 18, our method attributes the patient's acute pruritic rash following intravenous ceftriaxone to IgE-independent mast cell activation, selecting E. This reflects an overemphasis on the rapid onset of symptoms while neglecting the underlying vascular mechanism. Our model incorrectly generalizes the timing to a non-IgE mast cell pathway, whereas the characteristic feature of immune-mediated dermal vascular hyperpermeability better explains the urticarial appearance. The correct answer, A. Hyperpermeability of the superficial dermal microvasculature, accounts for the localized erythematous macules and the typical pathophysiology of an acute drug eruption. This failure highlights a reasoning bias where temporal cues (rapid onset) is overweighted, while the specific dermatological morphology and mechanism are under-integrated into the final judgment.

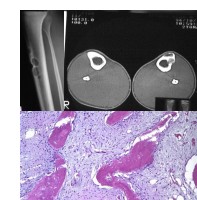 🧑‍⚕️ A 12-year-old girl presents with a prominent mass below her left knee that is tender upon impact but asymptomatic during rest and regular activities. She reports no fatigue, weight loss, or other masses. Physical examination reveals mild tenderness when the prominence is firmly palpated, with full, pain-free range of motion in both knee and ankle joints. Based on the provided radiograph (Figure A), CT scan (Figure B), and histological image (Figure C), what is the most probable diagnosis? **Options**: (A) Osteofibrous dysplasia (B) Low-grade chondrosarcoma (C) Aneurysmal bone cyst (D) Adamantinoma (E) Non-ossifying fibroma
*GT: A*

🤖 Lingshu-7B

<think>Based on the clinical presentation and imaging findings, we start by analyzing the radiograph and CT scan. The radiograph shows a well-defined, lytic lesion in the metaphysis of the tibia, which is characteristic of benign bone lesions. The CT scan confirms the presence of a multilocular cystic lesion with thin septations, typical of aneurysmal bone cysts. Histologically, the image shows fibrous tissue interspersed with blood-filled spaces, further supporting the diagnosis of aneurysmal bone cyst. This lesion is consistent with the patient's age and symptoms, as it is typically asymptomatic at rest and causes pain only upon impact. Other options such as osteofibrous dysplasia, low-grade chondrosarcoma, adamantinoma, and non-ossifying fibroma do not match the imaging and histological features as closely.</think> <answer>**C**</answer>

🤖 Ours-7B

Based on the image showing a prominent mass <finding> below the left knee <roi> that is tender upon impact <symptom> but asymptomatic during rest <fact>, this clinical presentation aligns well with the characteristics of osteofibrous dysplasia <hypothesis>. The imaging features reveal a lytic lesion with a \"soap bubble\" appearance <imaging_feature> on the CT scan, which is indicative of this benign bone tumor <support> <option> A.\n\nIn contrast, option B, low-grade chondrosarcoma, typically presents as a more aggressive lesion with significant pain and systemic symptoms such as fatigue and weight loss <fact>. The absence of these symptoms in the patient, along with the benign nature of the imaging findings, strongly refutes the diagnosis of low-grade chondrosarcoma <disprove> <option> B.\n\nSimilarly, for option C, aneurysmal bone cyst, while it can present with a lytic lesion, it usually exhibits more aggressive behavior and associated pain <fact>. The patient's asymptomatic status during regular activities and the benign imaging findings further support the diagnosis of osteofibrous dysplasia rather than an aneurysmal bone cyst <disprove> <option> C.\n\nRegarding option D, adamantinoma, this tumor is characterized by aggressive behavior and often presents with pain and systemic symptoms <fact>. The patient's lack of pain and the benign imaging findings do not support a diagnosis of adamantinoma <disprove> <option> D.\n\nLastly, option E, non-ossifying fibroma, typically does not present with significant tenderness or a prominent mass <fact>. The imaging findings of a lytic lesion with a \"soap bubble\" appearance are more characteristic of osteofibrous dysplasia, thus ruling out non-ossifying fibroma as a plausible diagnosis <disprove> <option> E.\n<answer>**A**</answer>

Figure 16: Comparison between Lingshu (Xu et al., 2025b) and Ours.

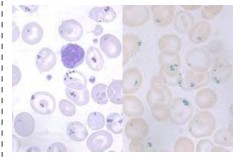 🧑‍⚕️ A 5-year-old boy is brought to the primary care clinic for his annual history and physical examination. His mother reports that he seems to tire more easily than other children his age and often lacks energy. He has been performing well in kindergarten and has met all developmental milestones. His medical history is unremarkable, and he is not on any daily medications. Family history includes two healthy older sisters and anemia in a maternal uncle. On examination, his temperature is 98.7\u00b0F, blood pressure is 108/66 mmHg, pulse is 130/min, and respiratory rate is 24/min. His height and weight are in the 15th percentile for his age. Mucosal pallor is noted on inspection. Laboratory tests show a hemoglobin level of 6.7 g/dL, a mean corpuscular volume of 65 fL, and a red cell distribution width of 30. A peripheral blood smear stained with Giemsa (left) and Prussian blue (right) is shown in *Image One.* Which of the following best characterizes the inclusions observed in this patient's erythrocytes? **Options**: (A) Iron (B) Ribosomes (C) Copper (D) Nuclear remnants (E) Denatured hemoglobin
*GT: A*

🤖 Chiron-o1-8B

Step 1: Analyze the clinical presentation and laboratory findings.\nThe patient is a 5-year-old boy with fatigue, mucosal pallor, and a significantly low hemoglobin level of 6.7 g/dL. His mean corpuscular volume (MCV) is markedly reduced at 65 fL, indicating microcytosis. The red cell distribution width (RDW) is also elevated at 30%, suggesting heterogeneity in red blood cell size. These findings are consistent with anemia of chronic disease or iron deficiency anemia.\n\n### Step 2: Interpret the peripheral blood smear findings.\nThe Giemsa-stained smear (left image) shows numerous erythrocytes with blue granular inclusions. The Prussian blue-stained smear (right image) reveals dark blue-black granules within the erythrocytes. Prussian blue staining is specific for iron deposition. Therefore, the blue granules observed in the Giemsa-stained smear are iron deposits.\n\n### Step 3: Correlate the findings with the differential diagnosis.\nThe patient has microcytic anemia with iron deposits in the erythrocytes. This pattern is characteristic of anemia of chronic disease, where iron is present but not utilized effectively due to inflammation or other underlying conditions. The presence of iron deposits in the erythrocytes is best described as sideroblastic anemia, a type of anemia of chronic disease.\n\n### Step 4: Determine the nature of the inclusions.\nThe Prussian blue stain confirms that the inclusions are iron. The Giemsa stain shows these iron deposits as blue granules within the erythrocytes. This confirms that the inclusions are iron deposits, not ribosomes, copper, nuclear remnants, or denatured hemoglobin.\n\n### The final answer is: (**A**) Iron

🤖 Ours-7B

Based on the image, the presence of <finding> basophilic stippling </finding> in the peripheral blood smear indicates the accumulation of <fact> ribosomal RNA </fact> within the red blood cells, which is characteristic of <hypothesis> ribosomes </hypothesis>. This finding aligns with the clinical presentation of <symptom> fatigue </symptom> and <symptom> lack of energy </symptom>, suggesting a potential underlying issue related to <hypothesis> ribosomal dysfunction </hypothesis>. Therefore, option <option> B. Ribosomes </option> is supported as it accurately describes the inclusions observed in this patient's erythrocytes.\n\nIn contrast, option <option> A. Iron </option> is incorrect because the presence of basophilic stippling does not correlate with iron accumulation; rather, it indicates ribosomal RNA accumulation. The laboratory findings of <finding> low hemoglobin level </finding> and <finding> low mean corpuscular volume </finding> further support the diagnosis of microcytic anemia due to ribosomal dysfunction, thus <refute> option A.\n\nSimilarly, option <option> C. Copper </option> is ruled out since the inclusions observed do not indicate copper accumulation but rather suggest ribosomal RNA accumulation. The clinical context of <symptom> fatigue </symptom> and <symptom> lack of energy </symptom> points towards ribosomal dysfunction rather than copper-related issues, thereby <refute> option C.\n\nOption <option> D. Nuclear remnants </option> is also incorrect as the basophilic stippling seen in the image is specifically associated with ribosomal RNA accumulation, not nuclear remnants. The laboratory results indicating <finding> low hemoglobin level </finding> and <finding> low mean corpuscular volume </finding> reinforce the diagnosis of microcytic anemia linked to ribosomal dysfunction, thus <refute> option D.\n\nLastly, option <option> E. Denatured hemoglobin </option> is not applicable here, as denatured hemoglobin would present differently on a blood smear and is not consistent with the findings of basophilic stippling. The clinical presentation and laboratory results suggest ribosomal dysfunction rather than denatured hemoglobin, leading to <refute> option E.\n<answer>**B**</answer>

Figure 17: Comparison between Chiron-o1 (Sun et al., 2025a) and Ours.

## A.10    Survey for Human Evaluation

To systematically assess the quality of our constructed SFT data and our model outputs, we design a targeted survey focusing on the use and logical consistency of structured reasoning. The survey includes three key evaluation dimension: (1) **factuality**, which examines whether the use of structured nodes correctly reflects the underlying content; (2) **consistency**, which evaluates the logical

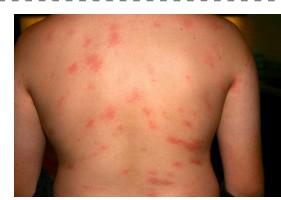

🧑‍⚕️💬 A 21-year-old woman presents with an itchy rash that appeared 2 hours after receiving an injection. Earlier today, she was diagnosed with uncomplicated pyelonephritis and treated with an intravenous dose of ceftriaxone, along with a prescription for oral ciprofloxacin. A photograph of the rash is provided. What is the most likely cause of this patient's skin findings? **Options**: (A) Hyperpermeability of the superficial dermal microvasculature (B) Tissue damage caused by the photochemical derivative of an antibiotic (C) Perivascular cutaneous polymorphonuclear infiltration (D) Cytotoxic T-cell destruction of antigen-tagged cells (E) IgE-independent mast cell activation in subcutaneous tissue
***GT: A***

🤖 Lingshu-7B

<think>Upon examining the image, the rash appears as erythematous macules and papules distributed across the back, which are consistent with a drug reaction. The timing of the rash onset shortly after receiving an intravenous injection suggests a hypersensitivity reaction to the administered medication. Given the history of treatment with ceftriaxone and ciprofloxacin, the most likely culprit is ceftriaxone, as it is known to cause allergic reactions including rashes. The mechanism of such reactions typically involves immune-mediated processes. Option A describes hyperpermeability of superficial dermal microvasculature, which aligns with the typical presentation of a drug-induced rash due to increased vascular permeability triggered by immune responses. This reasoning leads to the conclusion that the rash is most likely caused by hyperpermeability of the superficial dermal microvasculature.</think> <answer>**A**</answer>

🤖 Ours-7B

Based on the image showing an <finding> itchy rash that developed shortly after receiving an <fact> intravenous dose of ceftriaxone, it is evident that the timing of the rash aligns with the administration of the medication. This rapid onset suggests a direct reaction to the antibiotic rather than a delayed response, supporting option <option> E. IgE-independent mast cell activation in subcutaneous tissue </option>. The mechanism behind this type of reaction involves mast cells releasing histamine, leading to symptoms such as itching and hives, which matches the clinical presentation observed in this patient.\n\nIn contrast, option <option> A. Hyperpermeability of the superficial dermal microvasculature </option> does not adequately explain the immediate nature of the rash following the injection. While hyperpermeability can lead to skin reactions, it typically results in more diffuse symptoms rather than the localized rash seen here, thus <refute> option A.\n\nSimilarly, option <option> B. Tissue damage caused by the photochemical derivative of an antibiotic </option> is ruled out because there is no indication of sun exposure or UV light involvement in the patient's history. The absence of any mention of sunlight exposure further supports that this rash is not due to a photochemical reaction but rather a direct effect from the antibiotics administered, thereby <refute> option B.\n\nOption <option> C. Perivascular cutaneous polymorphonuclear infiltration </option> is also unlikely since this condition usually presents with more severe systemic symptoms and is associated with bacterial infections, which do not correlate with the patient's current diagnosis of uncomplicated pyelonephritis. The localized nature of the rash following the intravenous injection indicates a different mechanism at play, thus <refute> option C.\n\nLastly, option <option> D. Cytotoxic T-cell destruction of antigen-tagged cells </option> is not applicable here as cytotoxic T-cell responses typically manifest later and are associated with viral infections or malignancies, rather than the immediate reaction observed post-injection. Therefore, this option does not fit the clinical scenario presented, leading to <refute> option D.\n<answer>**E**</answer>

Figure 18: Comparison between Lingshu (Xu et al., 2025b) and Ours.

coherence among multiple nodes via constructed edges within the output; (3) **justification**, which assesses whether the structured output provides sufficient reasoning to support the model's conclusions. Specifically, human medical experts are asked to carefully annotate each dimension with a binary response (Yes/No) for the following questions: (1) Are the structured nodes accurately representing the relevant medical entity ? (2) Are the logical relationships of edges among nodes coherent ? (3) Does the structured reasoning sufficiently justify the model's conclusion ?

**SFT Data Verification:** we randomly sample 200 cases from our constructed SFT data. The human verification results, summarized in Table 11. Each entry represents the number of "Yes" responses out of 200 samples for each dimension. This table shows a pass rate above $98\%$ across all three dimensions, indicating that the majority of the constructed reasoning paths are factually accurate and logically coherent. While a full human verification of the entire dataset is infeasible, this sample provides strong evidence of the overall quality of the SFT data.

**Model Output Evaluation:** To rigorously assess the reliability of the structured reasoning produced by our model, we adopt a two-stage human evaluation protocol involving board-certified medical experts. (i) *Blind Correctness Assessment*: We sample 40 VQA examples (MedX-M (Zuo et al., 2025) and MMMU-Med (Yue et al., 2024)) and 40 QA examples (MedX-T (Zuo et al., 2025) and MMLU-Pro (Wang et al., 2024)) generated by PathFinder-7B. Each benchmark contributes an equal number of correct and incorrect model predictions, yielding 40 correct and 40 incorrect cases overall. These examples are randomly mixed, and the ground-truth answers are concealed. The medical expert is asked to independently judge whether the final answer is clinically correct based solely on the model's structured reasoning output. This blind evaluation eliminates label leakage and ensures that clinician judgments reflect the intrinsic interpretability and clinical validity of the reasoning process. Table 12 reports the results. The clinician correctly identifies 38 out of 40 correct predictions and 40 out of 40 incorrect predictions, demonstrating that the structured reasoning produced by PathFinder-7B provides sufficient clinical cues for experts to accurately assess correct-

ness. (ii) *Reasoning Quality Assessment*: We further select the 40 correctly predicted samples from PathFinder-7B (10 samples from each dataset, MedX-M, MMMU-Med, MedX-T and MMLU-Pro), and ask the expert to further evaluate the structured reasoning along the three dimensions: factuality, consistency and justification. The results are summarized in Table 13. Each entry represents the number of "Yes" responses out of 10 samples for each dimension, with higher scores indicating better performance. From the results, we observe that PathFinder-7B demonstrates high factual accuracy and logical consistency across both VQA and QA datasets, with average scores above 95.0% in all three dimensions. Notably, these results indicate that the model not only generates correct structured nodes but also organizes them into coherent reasoning chains that adequately support its conclusions. These findings provide further evidence of the reliability and interpretability of the model's structured outputs, complementing the quantitative performance metrics reported in the main paper.

Table 11: Survey of human evaluation of SFT data.

| Dimension | Factuality | Consistency | Justification | Average |
|---|---|---|---|---|
| SFT Data | 198/200 | 196/200 | 195/200 | 98.2% |

Table 12: Blind correctness assessment of model output.

| Model \ Human | TRUE | FALSE |
|---|---|---|
| Correct (40) | 38 | 2 |
| Incorrect (40) | 0 | 40 |

Table 13: Reasoning quality assessment of model output.

| Dimension | MedX-M | MMMU-Med | MedX-T | MMLU-Pro | Average |
|---|---|---|---|---|---|
| Factuality | 9/10 | 10/10 | 10/10 | 10/10 | 97.5% |
| Consistency | 9/10 | 10/10 | 9/10 | 10/10 | 95.0% |
| Justification | 9/10 | 10/10 | 9/10 | 10/10 | 95.0% |

## A.11 LIMITATIONS

While PathFinder substantially improves step- and branch-wise reasoning, several limitations remain. First, the model still relies heavily on broad and highly specialized medical knowledge; uncommon pathologies or rare multimodal cues can lead to misinterpretation, as illustrated in Figures 17 and 18. Second, integrating cross-modal evidence remains challenging: the model can overemphasize a single modality or superficial features, occasionally failing to reconcile conflicting cues from multiple sources. Third, reasoning fidelity is contingent on the quality of the underlying CoT data and expert annotations; errors or omissions in the graph-structured supervision can propagate into model predictions. Finally, the current framework primarily focuses on structured VQA and may require adaptation for more open-ended clinical tasks, such as free-text report generation or complex longitudinal reasoning. These limitations highlight areas for future improvement, including enhanced multimodal integration, continual medical knowledge updating, and robust handling of rare or ambiguous cases.

