# OpenReview forum: "PathFinder: Graph-structured Reasoning for Medical Visual Question Answering"
_ICLR.cc/2026/Conference — Submitted to ICLR 2026_

### Official Review · Reviewer_3g12 · 2025-10-30

**Soundness:** 2
**Presentation:** 2
**Contribution:** 3
**Rating:** 4
**Confidence:** 3

**Summary:**

The paper proposes a framework called PathFinder, a graph-structured reasoning framework for medical visual question answering. It represents clinical entities and causal/evidential links as nodes and edges, defines two new metrics (step-wise and branch-wise exploration), and trains with a new RL variant Graph-GRPO using three rewards (step, branch, outcome accuracy).
The paper reports experiments on 7 multimodal and 7 text-only medical benchmarks, claiming consistent performance gains.

**Strengths:**

- Extensive benchmarks and ablation studies.
- The high-level architecture and reward structure are well described.
- Combining graph-structured CoT supervision with process-level rewards is potentially impactful.

**Weaknesses:**

- Introduced Step and Branch evaluation metrics do not seem convincing to reflect the real impact of each method. Step counts the number of reasoning steps, branch counts if the reasoning steps are present for each option. However, throughout the paper it is difficult to derive the usefulness of these metrics. Although the authors mention the tendency of longer reasoning steps for correct answers in domain-specific VLMs and the opposite for general VLMs, it remains unclear whether such behavior is universal for all cases in all evaluated models or due to the underlying model architecture.
- Unclear which metric is reported in Tables 3 and 4.
- No evaluation for factuality, i.e., it remains uncertain whether the models generate factually correct reasoning. A human evaluation could have been beneficial in this regard.

**Questions:**

Refer to the weaknesses

---

> ### Author Response · Authors · 2025-11-21
> **Comments for Reviewer 3g12**
>
> We are thankful for your valuable feedback and acknowledgement of our work. Here are the responses to your concerns:
>
> **W1 Real impact**:
> (1) **Why Step/Branch are needed** in medical (V)QA, different reasoning failures exhibit several cases: shallow justification, premature commitment to single option, missing key clinical findings, failure to evaluate alternative hypotheses. Thus accuracy cannot reflect the structure of the reasoning.Step and Branch are explicitly designed to measure two fundamental dimensions of real clinical reasoning: Step-wise exploration in depth and casual layering of the reasoning, Branch-wise exploration in coverage of differential diagnoses. These two dimensions correspond directly to the actual diagnostic reasoning in clinical settings. (2) **Trends of existing VLMs**: Our experiments (Tab.1,2) show a consistent trend across all evaluated  4 medical and 4 general VLMs. This demonstrates that the metrics capture meaningful differences in reasoning quality and their impact is general rather than an artifact of a specific model architecture. (3) **Step/Branch are effective**: By integrating Step and Branch Rewards into our Graph-GRPO framework as process-level rewards, and ablation studies (Table 3–5) confirm that optimizing these rewards directly improves final answer accuracy. Additional results on open-ended medical report generation tasks (**[Common Question 1]**) further support that these metrics not only measure reasoning quality but also guide the model toward more effective and clinically-grounded diagnostic reasoning. (4) **Human verification confirmation**: Our human verification (in **[Common Question 2]**) confirms that the reasoning traces evaluated by these metrics are factually accurate, coherent, and well-justified.
>
> **W2 Unclear Metrics**: Table 3&4 report the final answer accuracy for QA or VQA tasks, following the evaluation protocol of each individual benchmark. Metrics are reported from the leaderboard in Lingshu [1], specifically indicating the percentage of choosing the correct choice out of candidate options.
>
> Reference: [1] Lingshu: A Generalist Foundation Model for Unified Multimodal Medical Understanding and Reasoning
>
> **W3 Human Evaluation**: We thank the reviewer for this suggestion. To assess the factual correctness and logical consistency of the model’s reasoning, we conduct two complementary forms of human evaluation,. (i) blind correctness assessment (ii) reasoning quality assessment. Please refer to **[Common Question 2]**, appendix A.10 and Table 11&12&13 for details.

---

> ### Author Response · Authors · 2025-11-28
> **Follow-up Comments for Reviewer 3g12**
>
> Dear Reviewer 3g12,
>
> Thank you again for the time and expertise you have devoted to reviewing our manuscript. We have posted point-by-point responses addressing all of your concerns. As the discussion period is approaching its end with less than a week remaining, we would like to ensure that we have fully resolved your concerns.
>
> If there are any remaining questions or points that you feel require further clarification, we would greatly appreciate your guidance. Your feedback is highly valuable to us, and we are committed to addressing any remaining issues to further improve the work.
>
> Thank you once again for your thoughtful review.

---

### Official Review · Reviewer_5aZu · 2025-10-31

**Soundness:** 2
**Presentation:** 3
**Contribution:** 2
**Rating:** 4
**Confidence:** 4

**Summary:**

The paper introduces PathFinder, a graph-structured reasoning framework for medical visual question answering (Med-VQA). Unlike traditional models that focus only on answer accuracy, PathFinder explicitly represents the reasoning process as a graph composed of medical entities such as symptoms, imaging features, hypotheses, and conclusions, connected by causal and evidential edges. It evaluates reasoning quality along two complementary dimensions: step-wise exploration, which measures causal coherence within the reasoning chain, and branch-wise exploration, which assesses the systematic exploration and elimination of diagnostic alternatives. The authors further propose Graph-GRPO (Graph-based Group Relative Policy Optimization), which integrates two structured rewards—step reward and branch reward—into reinforcement learning to encourage causally consistent reasoning and comprehensive diagnostic exploration. Experiments on seven multimodal and seven textual Med-VQA benchmarks demonstrate that PathFinder-7B achieves substantial improvements over existing models such as Lingshu-7B, Chiron-o1-8B, and MedVLM-R1-2B, offering enhanced clinical interpretability and structurally coherent reasoning paths.

**Strengths:**

1.	The paper formalizes the medical diagnostic process as a graph-structured reasoning paradigm, explicitly modeling entities and causal relationships to provide a systematic framework for interpretable reasoning in Med-VQA.
2.	Graph-GRPO jointly optimizes both step-wise and branch-wise dimensions, offering stronger process-level supervision and verifiability compared to traditional accuracy-only reinforcement learning approaches.
3.	Extensive experiments across 14 datasets demonstrate that PathFinder achieves state-of-the-art performance on both multimodal and textual Med-VQA tasks, while significantly enhancing reasoning depth and coverage.

**Weaknesses:**

1.	The construction of graph structures and reasoning evaluation relies on the LLM-as-a-Judge approach (GPT-4o), which may introduce bias and instability, lacking automated and reproducible validation.
2.	The graph generation and reasoning path construction require multiple iterative calls and manual verification, leading to high computational and human costs that hinder large-scale deployment.
3.	The performance may be sensitive to hyperparameter configurations, yet the paper does not provide systematic ablation studies or sensitivity analyses.
4.	Although the authors claim expert validation, no inter-rater consistency metrics or error case analyses are reported, leaving the clinical reliability insufficiently quantified.
5.	The paper lacks discussion of related studies on the application of Chain-of-Thought (CoT) reasoning in medical contexts, which could provide valuable comparative insights and theoretical grounding.

**Questions:**

1.	How do the authors ensure the consistency and correctness of each edge (causal or evidential) within the constructed graph? Are there potential issues of pseudo-causal chains or redundant edges?
2.	Do the Step and Branch Rewards in the Graph-GRPO framework conflict during optimization? How are these two objectives balanced to maintain stable learning?
3.	If PathFinder were applied to other medical modalities (e.g., multi-organ CT or pathology slides), would the graph node definitions and construction strategy require adaptation or re-design?
4.	Can the reliability of the LLM-as-a-Judge evaluation be quantitatively assessed? Have the authors considered measuring agreement between expert and model judgments using metrics such as Cohen’s κ?
5.	Since the current model performs single-round reasoning, could the framework be extended to multi-turn interactive diagnostic reasoning to better reflect clinical workflows?
6.	Could the graph structure be integrated directly into the visual encoding or Transformer layers rather than applied as a post-hoc reasoning stage?
7.	In real-world clinical scenarios, how can the system prevent the generation of misleading or overconfident explanations?
8.	Compared with existing paradigms like structured reasoning and process supervision, does PathFinder demonstrate sufficient generalizability and adaptability across diverse medical tasks?

---

> ### Author Response · Authors · 2025-11-21
> **Comments for Reviewer 5aZu**
>
> We are thankful for your valuable feedback and acknowledgement of our work. In response to the your comments, we have added new analyses, including human–LLM agreement studies, correctness verification, stability and sensitivity ablations, expanded related works, and additional experiments on open-ended task, to strengthen the reliability, generalizability, and empirical grounding of our method.
>
> **W1&W4&Q4 LLM-as-Judge**:  We acknowledge your concerns on the reliability of automated LLM evaluation. We have conducted human evaluation with LLM-as-Judge, showing the consistency and alignment (Spearman ρ=0.846&0.822) with human judgement. (i) To ensure consistency, for each model in Tab1, we first randomly select 5 examples within each dataset and have them manually evaluated with the same metrics by a licensed medical expert. These expert judgments are then used as examples, provided in the prompts to guide the LLM-as-Judge. (ii) To quantitatively assess reliability, we further select 10 examples per model per dataset in Tab1, resulting in 360 samples in total (9 models × 4 datasets × 10 examples). We compare the LLM-as-Judge scores with the human expert scores and compute Spearman correlation coefficients. The results show strong agreement: Branch Exploration ρ = 0.846 and Step Exploration ρ = 0.822, demonstrating that the LLM-as-a-Judge evaluation aligns well with human judgments. The full results are now added in Table 6 and Appendix A.3.2.
>
> ### Table: Spearman correlation for BranchExploration
> ||MedX-M|MMMU|MedX-T|MMLU-Pro|Avg.|
> |-|-|-|-|-|-|
> |ρ-value|0.849|0.884|0.824|0.826|**0.846**|
>
> ### Table: Spearman correlation for StepExploration
> ||MedX-M|MMMU|MedX-T|MMLU-Pro|Avg.|
> |-|-|-|-|-|-|
> |ρ-value|0.818|0.858|0.813|0.801|**0.822**|
>
> **W2 Human Cost**: We appreciate the reviewer’s observation. We would like to clarify that the computational and human cost associated with data construction is not unique to our method, but a well-recognized bottleneck across recent medical reasoning VLM research. State-of-the-art systems also rely on extensive LLM calls and human curation during data construction. For example SOTA models: **InfiMed: 224K, Lingshu: 11M, Chiron-o1: 492K, MedVLThinker: 200K, while Ours: 114K**. These works similarly acknowledge the high cost of data generation, reflecting the fact that medical reasoning supervision is inherently expensive. We agree that reducing the cost is a critical direction for future work.
>
> **W3 Hyperparameter & Sensitivity**: (1) We would like to clarify that systematic ablation studies have already been conducted. Specifically, we compare(different stage of training): baseline, baseline + SFT, and baseline + SFT + RL (reported in Table 6&7 of the original submission or Table 8&9 in the rebuttal revised submission). In addition, we perform ablations for the two rewards, Step Reward and Branch Reward, reported in Table 5. All hyperparameter settings are detailed in Section 4.1 and Appendix A.5. (2) To further assess sensitivity, we conduct a new ablation on the model’s temperature parameter τ, testing values τ = 0.25, 0.50, 0.75 in addition to the original τ = 1.0. The results, now added in appendix A.8 and Table 10, show that PathFinder’s performance is stable across this range, indicating robustness to the temperature hyperparameter.
> ### Table: Sensitivity analysis
> |Models|τ=1|τ=0.75|τ=0.50|τ=0.25|Average|
> |-|-|-|-|-|-|
> |Baseline-7B|22.2|22.0|21.0|21.1|21.6±0.53|
> |+Cold-start|26.1|26.5|26.0|26.5|26.3±0.23|
> |+Graph-GRPO|28.2|28.4|26.5|28.5|27.9±0.82|
>
> **W5 Related Studies**: Thanks for your suggestion, We have now added an additional paragraph with new reference paper in the related works section.
>
> **Q1 Correctness & Q7 Misleading**: We thank the reviewer for raising these important questions regarding reasoning correctness and reliability in real-world clinical scenarios. As detailed in our [Common Question 2], we systematically assessed both the SFT data and model outputs through human evaluation (Appendix A.10, Tables 11–13). Specifically, we sampled 200 SFT data points and evaluated (i) factuality of structured nodes, (ii) logical consistency among edges, and (iii) sufficiency of justification, achieving an average pass rate above 98%, which provides strong evidence that the training graphs are generally accurate and coherent. For the model outputs, we conducted a two-stage expert evaluation: (i) a blind correctness assessment on 80 randomly mixed samples (40 correct / 40 incorrect) to verify that the structured reasoning alone enables accurate clinical judgment, and (ii) a reasoning quality assessment on the 40 correctly predicted cases along the same three dimensions, achieving >95% pass rates. These results demonstrate that PathFinder-7B produces factually grounded, coherent, and well-justified reasoning chains, providing clinicians with reliable cues.

---

> ### Author Response · Authors · 2025-11-21
> **Additional Comments for Reviewer 5aZu**
>
> We are thankful for your valuable feedback and acknowledgement of our work. Here are some additional comments.
>
> **Q2 Balanced Training**: We additionally visualize the training trajectories of Step and Branch Rewards in the revised submission PDF. As shown in Figure 12 (Appendix), both rewards steadily increase alongside the Accuracy Reward, indicating that multi-objective optimization is stable. From a design perspective, the two rewards target complementary aspects of reasoning: Step Reward encourages finer-grained, sequential reasoning and clinically grounded steps, while Branch Reward promotes coverage across alternative candidate hypotheses. Since step reward and branch reward are orthogonal, there is no conflicts. Empirically, as reported in Table 5, using either reward alone improves performance, and combining both consistently yields the best overall results, confirming that our multi-objective design effectively balances the two objectives while enhancing final diagnostic accuracy.
>
> **Q3&Q8 Other Modalities & Generalization**: The seven VQA datasets used in our experiments already cover multiple imaging modalities, including X-ray, CT, MRI (VQA-RAD, SLAKE), and pathological imaging (PathVQA), as well as multimodal clinical text-image datasets (PMC-VQA, OmniMedVQA, MMMU, MedXpertQA) with multiple categories. Our graph node definitions and construction strategy have been applied consistently across these diverse modalities, demonstrating their flexibility and generalization capability. Furthermore, we have extended PathFinder to medical report generation as an open-end task.  Please refer to [Common Question 1] for details.
>
> **Q5 Multi-Round**: We primarily evaluate our model in a single-round reasoning setting to ensure a fair comparison with existing reasoning models. Nonetheless, extending the framework to multi-turn, interactive diagnostic reasoning is a direction for future work. Such an extension could more closely emulate real-world clinical workflows and may involve integrating agentic reasoning paradigms such as ReAct, multi-turn dialogue exploration, or retrieval-augmented generation (RAG) strategies.
>
> **Q6 Encoding Layer**: We thank the reviewer for this insightful suggestion. Integrating graph structure directly into the visual encoder or Transformer layers could be a future direction. We clarify that the main focus of PathFinder is not architectural modification of the backbone, but the development of a model-agnostic structured reasoning framework. This design intentionally keeps the graph post-hoc and detachable, allowing it to be applied across different VLMs without changing their internal architecture.

---

> ### Author Response · Authors · 2025-11-28
> **Follow-up Comments for Reviewer 5aZu**
>
> Dear Reviewer 5aZu,
>
> Thank you again for the time and expertise you have devoted to reviewing our manuscript. We have posted point-by-point responses addressing all of your concerns. As the discussion period is approaching its end **with less than a week remaining**, we would like to ensure that we have fully resolved your concerns.
>
> If there are any remaining questions or points that you feel require further clarification, we would greatly appreciate your guidance. Your feedback is highly valuable to us, and we are committed to addressing any remaining issues to further improve the work.
>
> Thank you once again for your thoughtful review.

---

### Official Review · Reviewer_smHP · 2025-11-06

**Soundness:** 3
**Presentation:** 3
**Contribution:** 3
**Rating:** 6
**Confidence:** 4

**Summary:**

This paper proposes PathFinder, a graph-structured reasoning framework designed to enhance the reasoning capability of medical VLMs in medical VQA tasks. The authors argue that existing VLMs often suffer from step-wise and branch-wise deficiencies—i.e., their reasoning chains are incomplete, non-causal, or fail to systematically explore alternative diagnostic routes. To address this, the paper introduces Graph-GRPO, which encourages both step-wise and branch-wise exploration by incorporating graph-structured supervision along with Step Reward, Branch Reward, and outcome accuracy.
However, the step and branch rewards only encourage more logical and comprehensive reasoning, without explicitly evaluating the correctness of the reasoning process. In my view, this is the major weakness of the current framework and an important direction for future improvement. Extensive experiments on multiple multimodal and text-only benchmarks demonstrate that PathFinder achieves superior accuracy and produces more structured and comprehensive reasoning chains than existing models.

**Strengths:**

1. The proposed Graph-GRPO effectively enforces a more causal, logical, and comprehensive reasoning process through graph-structured supervision.

2. The PathFinder-7B model trained with this strategy achieves better performance than existing medical VLMs across multiple benchmarks.

**Weaknesses:**

1. The Step Reward and Branch Reward primarily encourage the model to produce logically coherent and comprehensive reasoning but do not evaluate the correctness or factual validity of the reasoning process.

2. The construction of the reasoning graph is a key component of the framework, yet the definitions of nodes and edges are somewhat vague. Specifically, nodes are defined as medical findings, clinical symptoms, hypotheses, rules, evidence, and imaging features or regions of interest, while edges represent support, derive, refute, or rule out relations. The rationale behind these specific categories is not clearly justified—for instance, why are symptoms and imaging features not considered subtypes of findings? And why are the edge types limited to these four relationships?

3. The proposed approach is tailored for multiple-choice or closed-ended medical VQA tasks. Its applicability to open-ended VQA tasks appears limited.

**Questions:**

1. Could the authors further clarify the taxonomy of nodes and edges in the reasoning graph? What criteria were used to define and separate these categories?

2. How could the proposed framework be adapted for open-ended medical QA or report generation tasks, where no predefined candidate answers exist?

3. Have the authors considered incorporating a reasoning correctness or factual consistency reward, in addition to the step and branch rewards, to more directly guide the model toward clinically valid reasoning paths?

---

> ### Author Response · Authors · 2025-11-21
> **Comments for Reviewer smHP**
>
> We are thankful for your valuable feedback and acknowledgement of our work. In response to the your comments, we clarify the intentional design of structural rewards, additional verification of reasoning trace, provide a detailed rationale for the node and edge taxonomy grounded in clinical diagnostic practice, and extend our evaluation to open-ended medical tasks while discussing the challenges of step-level factuality rewards.
>
> **W1 Reward Design:** We agree that the Step Reward and Branch Reward emphasize the structural quality of reasoning rather than factual correctness. This design is intentional. Our goal is to encourage the model to generate clinically interpretable and well-organized reasoning traces, which provide a stable scaffold for medical decision-making. The factual correctness of reasoning is enforced separately through the accuracy reward.
> Importantly, our ablation studies show that introducing structural rewards consistently improves final diagnostic accuracy. This indicates that better-organized and clinically-grounded reasoning chains provide more reliable cues for the model to reach correct medical conclusions. In other words, Step/Branch rewards and the correctness reward are complementary. For the correctness of reasoning, we conduct additional human verification, please refer to [Common Question 2(2)(ii)].
>
> **W2&Q1 Nodes & Edges Design:** The node and edge taxonomy used in PathFinder is not arbitrary; it is derived from how clinical diagnosis is structured in real medical practice. Although symptoms, findings, and imaging features may all seem like “observations,” they play different epistemic roles in clinical decision-making: (1) Symptoms are subjective patient-reported observations. (2) Findings are objective clinician-confirmed observations. (3) Imaging features / ROIs are a specialized subclass of findings that require spatial grounding and serve as key anchors for imaging reasoning. (4) Hypotheses represent diagnostic candidates under consideration. (5) rules / evidence encode medical knowledge used to connect observations to hypotheses. These categories mirror the sequential stages of clinical diagnosis, from subjective complaint to objective examination, rule-based inference, and hypothesis evaluation. In addition to these types stated in the main paper, we have included top-10 most frequent node and edge categories in our constructed graphs, as shown in Figure 10(a)(b) in the appendix.
>
> **W3&Q2 Open-End Task**: We additionally conduct experiments on open-end medical report generation task. Please refer to the [Common Question 1].
>
> **Q3 Process Reward**: Designing a reliable step-level correctness signal in medical reasoning remains an open challenge: intermediate reasoning steps, detailed medical expertise, and current automated factuality evaluators (such as Process Reward Model) are still unreliable. As a result, incorporating such a reward risks introducing noise or bias. We therefore consider this an important bottleneck and a promising direction for future research.

---

> > ### Comment · Reviewer_smHP · 2025-11-26
> > **Remaining Concerns on Edge Taxonomy and Open-Ended Adaptation**
> >
> > Thank you for the responses. While the motivation behind Step and Branch Rewards is clear and reasonable, two concerns remain insufficiently addressed.
> >
> > (1) The edge taxonomy still lacks a principled or systematic justification. The response provides clinical intuition but no clear methodology for defining and constraining the set of edge types.
> >
> > (2) The adaptation to open-ended tasks is not fully explained. Providing evaluation results does not clarify how graphs and rewards are constructed when no candidate answers exist, nor whether additional mechanisms (e.g., self-generated hypotheses) are required.

---

> > > ### Author Response · Authors · 2025-11-28
> > > **Response to the remaining concerns**
> > >
> > > Thank you for the feedback and engagement in the discussion !
> > >
> > > 1. **Edge taxonomy**: We further explain and clarify both the principle and the procedure behind the design of edge.
> > >
> > > (1) **principle**: Our goal is not to enumerate all possible linguistic relations, but to capture the minimal set of patterns that clinicians routinely use to relate image findings, intermediate hypotheses, diagnostic conclusions and other medical entities. Importantly, the taxonomy of edges is not hand-crafted in advance, but rather it emerges from how clinicians naturally connect these node types through causal and evidential reasoning. In clinical practice, relations between entities fall into four fundamental diagnostic actions: (i) support: evidence increases the plausibility of an entity (ii) derive: one entity follows from another via a causal or logical rule (iii) refute: one medical entity contradicts the other entity (iv) rule out: criteria excludes an entity from the differential diagnosis.
> > > **For example**, (a) the painful toe discoloration together with a pulsatile abdominal mass (findings) allows clinicians to (derive) the (hypothesis) of the related emboli via established vascular (evidence), … and these ischemic (findings) further (support) the diagnosis of fibrinous thrombi. (b) Conversely, the mismatch between expected (evidence) and observed (findings) enables clinicians to (invalidate) alternatives: the absence of vascular or metabolic abnormalities (refutes) options such as … diseases such as pericarditis or non-ossifying fibroma can be (ruled out) because essential (imaging_feature) are not present.
> > > Every relation, including positive, causal, contradictory, or eliminative, naturally maps to one of the four edge types, linking medical entity nodes. This demonstrates that the edge taxonomy arises from real diagnostic reasoning and provides a clinically verifiable structure for explaining the decision.
> > >
> > > (2) **procedure**: In the construction of the reasoning graph, we do not pre-assign one of the four labels directly. Instead, GPT-4o are prompted to produce proper free-form natural language descriptions of how two entities are related. To verify whether the four-type taxonomy sufficiently captures real-world relations, we sample 200 SFT examples, yielding 976 raw relation phrases. We manually group these raw relation edges into clusters. We find that the relations fall into four semantic clusters corresponding to support, derive, refute, rule out. The distribution is
> > > ### Table: Distribution of Edges
> > > ||Support|Derive|Refute|Rule out|
> > > |-|-|-|-|-|
> > > |**SFT Data** |134/976|298/976|442/976|102/976|
> > >
> > > (each data sample contain multiple edges presenting the relation between nodes)
> > > Additionally, we verify the logical consistency of edges, as shown in previous **[Common Question 2] Table: Survey of human evaluation of SFT data** with 196/200.  In summary, these edge types form a principled, clinically interpretable, and data-validated set of reasoning primitives that fully cover the relations required for diagnostic reasoning while keeping the model’s reasoning steps transparent and clinically verifiable.
> > >
> > > 2. **Detailed clarification on open-end task**: We adopt the same SFT$\rightarrow$Graph-GRPO training pipeline used for closed-ended tasks, with adaptations to graph construction and reward design. We randomly sample 4k data (2k for SFT and 2k for RL) from the training split of CheXpert Plus dataset for the medical report generation task.
> > >
> > > (1) Graph construction: in open-ended tasks, we construct the diagnostic graph using the same pipeline as in closed-ended VQA. Nodes are extracted and edges are generated exactly as in the closed-ended setting, capturing evidence, hypothesis relation, differential-diagnosis logic with medical entities.
> > >
> > > (2) SFT data generation: the diagnostic graph is unfolded into a reasoning trajectory that precedes the final report. The output format follows the template: **... 'Reasoning Process' ... < Final Report > Findings: …  Impression: … </Final Report>**. Although no explicit candidate answers exist, the reasoning path still includes exploration of alternative hypotheses and rule-out logic, preserving the same reasoning structure.
> > >
> > > (3) Graph-GRPO adaptation: Process-level rewards (Step Reward and Branch Reward) remain unchanged. The outcome reward is replaced by a sentence-level semantic reward, BertScore-F1, continuous value between 0.0 and 1.0. The final reward is
> > > $r=\lambda_{\text{step}} r_{\text{step}}  + \lambda_{\text{branch}} r_{\text{branch}} + \lambda_{\text{acc}} BERTScore-{F1}(y,\hat{y}) $ where $y,\hat{y}$ correspond to the ground truth and predicted final report. These hyperparameters $\lambda_{\text{step}},r_{\text{step}}, \lambda_{\text{branch}},r_{\text{branch}},\lambda_{\text{acc}}$ are consistent with close-end task defined in Section3.3.
> > >
> > > We have now added these details on A.6 in the appendix. If you still have further concerns, we are committed to address.

---

### Author Response · Authors · 2025-11-21
**Common Questions to all Reviewers**

We are thankful to all reviewers for providing valuable feedbacks and insights of our submission. Here are some common questions:

**1. Open-End Task**

In addition to the conventional QA and VQA benchmarks, we further evaluate our model PathFinder’s performance on an open-end task: medical report generation. The results are summarized in Table7 in the appendix, including two widely adopted benchmarks: CheXpert Plus, and IU-Xray. We follow the leaderboard in Lingshu and their evaluation codes and specifically report two semantic-based metrics ROUGE-L and CIDEr, one model-based metrics RaTE. Our PathFinder-7B model achieves competitive or superior results compared with both state-of-the-art open-source and proprietary models. Together with Table 3&4, these results show that our model superior performance in both close-end (V)QA and open-end medical report generation task. We provide a short table here. For the full table, please refer to Table 7 in the appendix, with an example in Figure 11.

### Table: Medical report generation tasks (CXP: CheXpert Plus and IU-Xray)

| Model | ROUGE-L (CXP) | CIDEr (CXP) | RaTE (CXP) | ROUGE-L (IU-Xray) | CIDEr (IU-Xray) | RaTE (IU-Xray) |
|-|-|-|-|-|-|-|
| GPT-4.1 | 24.5 | 78.8 | 45.5 | 30.2 | 124.6| 51.3 |
| MedGemma-4B-IT| **27.1** | _79.0_| **47.2** | 30.8| 103.6| 57.0|
| LLaVA-Med-7B| 18.4 | 45.5| 38.8| 18.8  | 68.2  | 40.9  |
| HuatuoGPT-V-7B| 21.3 | 64.7| 44.2 | 29.6 | 104.3 | 52.9 |
| Qwen2.5VL-7B| 22.2| 62.0 | 41.0| 26.5 | 78.1| 48.4 |
| Lingshu-7B | _26.5_  | _79.0_ | 45.4| **41.2** | **180.7**| _57.6_ |
| *PathFinder-7B (Ours)* | 26.1| **88.7** | _46.9_ | _35.0_ | _153.6_  | **63.5**  |

**2. Reasoning Correctness:**

We thank the reviewer for raising the concern regarding factual correctness and reliability of the reasoning data. To address this, we conducted a comprehensive human evaluation with a licensed medical expert, covering both the constructed SFT data and the model’s structured reasoning outputs. For detailed explanation, please refer to appendix A.10 and Table 11&12&13 in the revised submission PDF.

(1) **SFT data verification**: We randomly sample 200 SFT data examples and ask the medical expert to evaluate three key dimensions: (i) factuality of structured nodes, (ii) logical consistency of edges, (iii) sufficiency of justification for the final conclusion. As shown in Table below, the pass rate exceeded 98% across all dimensions, indicating that the constructed reasoning graphs are accurate and coherent. While full manual verification of 100k samples is infeasible, this sample provides strong evidence for overall data quality.

### Table: Survey of human evaluation of SFT data

| Dimension   | Factuality | Consistency | Justification | Average |
|-|-|--|-|-|
| SFT Data | 198/200| 196/200| 195/200| 98.2%|

(2) **Model output evaluation**: We further performed a two-stage evaluation of PathFinder-7B’s outputs in two way:

(i) **Blind Correctness Assessment**: We sample 80 model outputs (40 correct / 40 incorrect) from 2 VQA (MedX-M, MMMU-Med) and 2 QA (MedX-T, MMLU-Pro) datasets. The cases are mixed and the ground truth label is withheld. The expert judges the correctness solely based on the structured reasoning without knowing the ground truth label. As summarized in Table below, clinicians correctly identified 38/40 correct and 40/40 incorrect predictions, demonstrating that the model’s reasoning traces contain sufficient clinical signals for expert to reliably assess correctness.

### Table: Blind correctness assessment of model output

| Model \ Human | TRUE | FALSE |
|-|-|-|
| Correct (40)|38|2|
| Incorrect (40)|0|40|

(ii) **Reasoning Quality Assessment**: We then evaluate the 40 correctly predicted cases (10 per dataset from MedX-M, MMMU-Med, MedX-T, MMLU-Pro) along factuality, consistency, and justification. Results show that the model achieves >95% pass rate, confirming that PathFinder-7B produces factually grounded, logically coherent, and well-justified reasoning chains.

### Table: Reasoning quality assessment of model output

| Dimension| MedX-M | MMMU-Med | MedX-T | MMLU-Pro | Average |
|-|-|-|-|-|-|
| Factuality| 9/10| 10/10| 10/10| 10/10| 97.5%|
| Consistency| 9/10| 10/10| 9/10| 10/10| 95.0%|
| Justification| 9/10| 10/10| 9/10| 10/10| 95.0%|

---

### Author Response · Authors · 2025-12-03
**Rebuttal Summary**

We sincerely thank the AC and reviewers for their feedback. We summarize this paper and how we have resolved the concerns from the reviewers.

**Paper overview:**

We propose PathFinder, a graph-structured reasoning framework for Medical VQA. Medical entities (symptoms, findings, imaging features/ROIs, hypotheses, rules/evidence, conclusions) are represented as nodes, and causal/evidential relations (support, derive, refute, rule out) as edges. On top of this representation, we formalize two structural dimensions of reasoning quality: Step-Wise Exploration (depth and causal layering of intermediate steps) and Branch-Wise Exploration (coverage and elimination of alternative hypotheses with extensive experiments on existing VLMs. We further introduce Graph-GRPO, a reinforcement learning scheme with three rewards: (i) Step Reward encouraging causally coherent step-wise reasoning, (ii) Branch Reward encouraging systematic exploration/rule-out of options, and (iii) outcome accuracy accuracy. Using Qwen2.5-VL-7B as the backbone, PathFinder-7B is evaluated on 7 multimodal and 7 text-only medical benchmarks and achieves state-of-the-art performance compared with recent medical reasoning VLMs (e.g., Lingshu-7B, Chiron-o1-8B). The framework is also extended to open-ended medical report generation (CheXpert Plus, IU-Xray) with competitive or superior performance.

During the rebuttal period, we have revised the manuscript to address the following concerns:

**Main concerns from reviewers and responses:**

1. Nodes/Edges taxonomy and justification (Reviewer smHP):

Nodes are mapped to distinct roles in clinical diagnosis (symptoms, findings, imaging features / ROIs, hypotheses, rules / evidence). Edges correspond to four core diagnostic actions: increasing plausibility, causal derivation, contradiction, and exclusion from the differential. We further sampled 200 SFT examples (976 raw relations) and manually clustered LLM-generated relation phrases into these four groups (134 / 298 / 442 / 102), suggesting that the taxonomy is minimal, clinically interpretable, and empirically sufficient.

2. Open-ended tasks (Reviewer smHP & 5aZu):

The same graph construction and SFT → RL pipeline is applied on CheXpert Plus (2k SFT, 2k RL). Nodes/edges are built as in closed-ended VQA. The graph is unfolded into “Reasoning Process + Final Report (Findings / Impression)”. Step/Branch rewards remain unchanged; the accuracy reward is replaced by a sentence-level BERTScore-F1 between generated and reference reports. PathFinder-7B achieves strong ROUGE-L/CIDEr/RaTE on CheXpert Plus and IU-Xray.

3. Usefulness of Step/Branch metrics (Reviewer 3g12):

StepExploration measures depth/causal layering; BranchExploration measures coverage of differential diagnoses. Across 4 medical and 4 general VLMs, consistent patterns are observed as described in paper Section 3.1. Ablations show that adding Step/Branch rewards on top of SFT + accuracy reward improves final accuracy, indicating these metrics are useful both as evaluation indicators and learning signals.

4. Factuality and LLM-as-Judge reliability (Reviewers 5aZu & 3g12):

We conduct several human evaluations (i) LLM–Human agreement on 360 samples, reporting high Spearman correlations for Step/Branch (ρ≈0.82–0.85); (ii) SFT graph quality on 200 samples, with >98% pass rate on node factuality, edge consistency, and justification; and (iii) model output evaluation with 78/80 accuracy in a blind correctness test and >95% pass rates in reasoning quality on correct predictions. These results suggest that both the training data and PathFinder’s outputs are clinically coherent and factually reliable.

5. Cost, sensitivity, and generalization (Reviewer 5aZu):

We note that our data scale (~114k examples) is moderate compared to other medical reasoning VLMs (InfiMed: 224K, Lingshu: 11M, Chiron-o1: 492K, MedVLThinker: 200K). We report ablations over training stages and reward components, and add a temperature sensitivity study (τ=1.0, 0.75, 0.5, 0.25) showing stable performance. The same node/edge and graph construction are applied across multiple imaging modalities (X-ray, CT, MRI, pathology) and text-only tasks, and the approach is also validated on open-ended report generation. Extensions to multi-turn reasoning and encoder-integrated graph architectures are discussed as future work.

---

### Meta-Review · Area_Chair_Kvwf · 2026-01-02

**Summary:**

Medical Visual Question Answering (MVQA) aims to improve interpretability and clinician confidence by presenting explicit, evidence-based reasoning alongside diagnostic predictions. However, existing methods often provide incomplete, non-causal explanations that frequently overlook critical evidence, intermediate steps, and alternative hypotheses. Therefore, this paper proposes "PathFinder," a graph reasoning framework that represents medical entities as nodes and causal and evidence relationships as edges. PathFinder systematically explores diagnostic pathways. PathFinder introduces two types of exploration: (1) stepwise (searching for intermediate entities with causal links) and (2) branching (searching for alternative paths and eliminating unnecessary candidates). Furthermore, the paper introduces Graph-GRPO, which integrates graph structure supervision with two types of process rewards: step rewards and branch rewards.

 This work introduces graph structure inference and step/branch metrics/rewards to medical VQA. Through rebuttal and additional experiments, many practical concerns were addressed to a certain extent (e.g., design explanation and additional evaluation). However, the Step/Branch metric remains insufficiently validated as an indicator of inference quality. The observed accuracy improvement could be explained by redundancy and increased search volume rather than actual quality enhancement. Second, the scalability response remains limited to general data creation principles and fails to address the design bottlenecks of the proposal directly: inference/operational costs and reliance on manual verification. Therefore, given the remaining risks regarding the substantiation of key claims and the practical feasibility, the AC recommends rejecting the proposal.

**Reviewer Concerns:**

Reviewers have raised the following concerns:

(1) Insufficient validity of the metrics (Step/Branch) and inference quality; (2) Ambiguous definition and rationale for the graph inference design; (3) Concerns about evaluation reproducibility and reliability; (4) Computational/human cost and scalability; (5) Insufficient analysis; and (6) Limited applicability.
The authors provided additional experiments and explanations in their rebuttal, which are believed to address many of the above concerns.

However, AC considers the following to be fundamental concerns that remain:

The response regarding the validity of the metrics (Step/Branch) and inference quality provides reasonable explanations for the necessity of shallow rationale, early commitment to a single option, and lack of differential diagnosis. The intuition behind dividing clinical reasoning into depth (Step) × comprehensiveness (Branch) is understandable as well. However, this primarily justifies the concept and does not directly address the reviewers' doubts about the validity of the metrics.
Furthermore, while increasing accuracy through Step/Branch is a strong counterargument, it is difficult to determine whether the reward effectively improved the intended "inference quality" or merely increased redundancy, cognitive load, or search effort. Consequently, it is difficult to conclude that these metrics capture the essence.

Regarding computational and human costs and scalability, it is undeniable that creating medical inference data is costly. However, this response to the reviewer's comment is insufficient. The reviewer's concern is not the general data creation cost in the medical field, but rather the scalability of the proposed method (graph generation and path construction) during "inference/operation" and the pipeline design itself, which relies on manual verification.

**Reviewer Scores:**

Reviewer smHP provided a score of 6 and participated in the discussion. However, it is unclear whether they were satisfied with the authors' final response.

Reviewers 5aZu and 3g12 gave a score of 4 but did not participate in the discussion. While the authors' rebuttal appears to address many of these reviewers' concerns, as stated in the Reviewer Concerns section, the concerns are not entirely resolved.

Therefore, it is questionable whether the two reviewers who gave negative scores will change them to a positive score of 5 or higher.

---

### Decision · Program_Chairs · 2026-01-26

Reject